# Towards a more reliable forecast of ice supersaturation: Concept of a one-moment ice cloud scheme that avoids saturation adjustment

Dario Sperber[1] and Klaus Gierens[1]

[1]Deutsches Zentrum für Luft- und Raumfahrt, Institut für Physik der Atmosphäre, Oberpfaffenhofen, Germany

**Correspondence:** Klaus Gierens (klaus.gierens@dlr.de)

**Abstract.** A significant share of aviation's climate impact is due to persistent contrails. Avoiding the creation of contrails that exert a warming impact is thus a crucial step in approaching the goal of sustainable air transportation. For this purpose, a reliable forecast of when and where persistent contrails are expected to form is needed, that is, a reliable prediction of ice supersaturation. With such a forecast at hand it would be possible to plan aircraft routes on which the formation of persistent contrails can be avoided. One problem on the way to these forecasts is the current systematic underestimation of the frequency and degree of ice supersaturation on cruise altitudes in numerical weather prediction due to the practice of "saturation adjustment". In this common parameterisation, the air inside cirrus clouds is assumed to be exactly at ice saturation, while measurement studies have found cirrus clouds to be quite often ~~in an ice supersaturated state~~ out of equilibrium.

In this study, we propose a new ice cloud scheme that overcomes saturation adjustment by explicitly modelling the decay of the in-cloud humidity after nucleation, thereby allowing for both in-cloud super- and subsaturation. To achieve this, we introduce the in-cloud humidity as a new prognostic variable and derive the humidity distribution in newly generated cloud parts from a stochastic box model that divides a model grid box into a large number of air parcels and treats them individually.

The new scheme is then tested against a parameterisation that uses saturation adjustment, where the stochastic box model serves as a benchmark. It is shown that saturation adjustment underestimates humidity both shortly after nucleation, when the actual cloud is still highly supersaturated, and also in aged cirrus if temperature keeps decreasing, as the actual cloud remains in a slightly supersaturated state in this case. The new parameterisation on the other hand closely follows the behaviour of the stochastic box model in any considered case. The improvement in comparison with saturation adjustment is largest if slow updraughts occur in relatively clean air in models with high spatial and temporal resolution. We conclude that our parameterisation is promising but needs further testing in more realistic frameworks.

## 1 Introduction

The upper troposphere is quite often in a state of ice supersaturation, both in clear air (e.g. Gierens et al., 1999; Petzold et al., 2020) and within cirrus clouds (Ovarlez et al., 2002; Spichtinger et al., 2004; Dekoutsidis et al., 2022). Although this state has been first reported in 1906 (Wegener, 1914), it was ignored in numerical weather prediction (NWP) until the end of the century, when it was first introduced into the U.K. Meteorological Office Unified Model (Wilson and Ballard, 1999) and later into the integrated forecast system (IFS) of the European Centre for Medium-Range Weather Forecasts (ECMWF, Tompkins et al.,

2007). However, NWP models that incorporate ice supersaturation in their cirrus parameterisations, often assume (at least, one-moment parameterisations) that supersaturation relaxes to saturation in the cloudy part of a grid box as soon as cloud formation occurs. Hence, this procedure that ~~is termed~~ represents a form of "saturation adjustment" (McDonald, 1963), but for ice clouds, ignores ice supersaturation within cirrus clouds. There is justification to do so, and the grid-mean water vapour mixing ratios are largely consistent with in-situ measurement results (at least in the IFS, Kaufmann et al., 2018). However, it leads to underestimation of the occurrence frequency and degree of ice supersaturation in upper-tropospheric layers (Gierens et al., 2020) and this turns out to be a problem for the forecasting of persistent contrails.

Aviation contributes about 3.5% of the total anthropogenic climate impact (Lee et al., 2021) through both $CO_2$ and non-$CO_2$ effects. In terms of radiative forcing and effective radiative forcing, the contrail-cirrus dominates among the non-$CO_2$ effects. The climate impact of persistent contrails is significant, but its climatological mean is quite uncertain. One source of this uncertainty is simply the vast weather-induced variability of the radiative effect of individual contrails (Wilhelm et al., 2021). Further difficulties arise from a lack of relative humidity measurements at cruise levels and from the mentioned underestimation of ice-supersaturation in current NWP models. The latter two issues also impede a reliable forecast of persistent contrails.

Contrails, like natural cirrus, scatter solar light and trap infrared radiation from Earth's surface and from lower atmospheric levels (e.g. Meerkötter et al., 1999; Corti and Peter, 2009; Schumann et al., 2012; Wolf et al., 2023), so they have both cooling and heating effects on the atmosphere, but, on average, the warming dominates (Stuber et al., 2006). The distribution of instantaneous radiative effects is wide (Wilhelm et al., 2021), and so is the distribution of contrail lifetimes (Gierens and Vázquez-Navarro, 2018), such that one can assume that the individual radiative effect of single contrails (energy forcing, see Schumann et al., 2012) varies widely as well. Thus, in order to drastically reduce the contrail share to aviation's climate effect, it suffices to avoid those contrails that exert the strongest warming impact (Teoh et al., 2020, 2022). In order to predict when and where contrails with strong warming impact can occur, one needs three steps: 1) Predict where contrails can form (Schmidt-Appleman criterion, see Schumann, 1996); 2) predict where they persist (ice supersaturated regions, ISSRs, see Gierens et al., 2012); and 3) determine or estimate their individual radiative effect either by simulating their development (Schumann, 2012) or by using so-called algorithmic climate change functions (aCCFs, see Yin et al., 2022; Dietmüller et al., 2022). The second of these steps is currently the bottleneck to a better contrail mitigation, because the prediction of ice supersaturation is challenging.

Satellite imagery taken in the water vapour absorption band at about 6 to 7 $\mu$m shows how variable the water vapour field is in the upper troposphere. Water vapour participates in dynamic, thermodynamic, chemical and aerosol processes on a multitude of spatial and time scales. Relative humidity changes due to variation in both water vapour concentration and temperature. This alone makes the prediction of the relative humidity field difficult. But moreover ice supersaturation is a ~~extremal~~ state of the humidity field that is far away from equilibrium making it more sensitive to changes of external conditions than bulk measures of humidity (e.g. mean concentration). As a consequence, there is a huge variability in upper tropospheric relative humidity, which renders in particular prediction of ice supersaturation with a precision (time, location) that suffices for environmentally friendly flight routing a serious challenge. This problem is aggravated by a lack of reliable humidity measurements in the upper troposphere that could be used in data assimilation for numerical weather prediction. There are long-term measurement

 It is clear that a field as variable as relative humidity needs a dense measurement network to be characterised reliably.

But this is not the topic of the current paper. As mentioned above, the current cirrus parameterisations used in NWP assume saturation adjustment and thus ignore ice supersaturation in cirrus clouds. We introduce a concept of a cirrus parameterisation that does not use saturation adjustment. Instead, we assume a simple distribution of supersaturation within cirrus and use the average in-cloud humidity as an additional prognostic variable. Furthermore, we acknowledge the fact that the in-cloud supersaturation relaxes to an equilibrium supersaturation a few percent above saturation as long as there is cooling

(Khvorostyanov and Sassen, 1998a). We develop the concept within a box model that represents one grid box of a 3D-model. The implementation of the new concept into a 3D-NWP model is left for future work.

The paper is organised as follows: the new concept is developed using a stochastic model, where the single grid box is divided into a large number of air parcels that have a certain initial distribution of relative humidities (or specific humidities). The results of this stochastic model are taken as the truth to check the new parameterisation which is developed in the next section as well.

The checks are presented in section 3, where we also compare the new parameterisation with one that closely follows the one used in the IFS, that is, one with saturation adjustment. The comparison makes the underestimation of supersaturation in the current models evident. We discuss some issues in section 4 and present our conclusions in the final section 5.

## 2   Stochastic box model and new parameterisation

### 2.1   Stochastic box model

We first want to explicitly model the evolution of humidity inside a model grid box that we consider to consist of an arbitrarily large number of air parcels. At the beginning of the simulation we set the grid box mean temperature $T$ to a start value below the threshold for spontaneous freezing of supercooled droplets, which is at about $-38°$C. We also set the cloud fraction $C$ to zero and thus the specific ice water content $q_i$ as well. We follow the approach selected for the ECMWF IFS and set the temperature in all parcels to a common value, $T$, and let the specific humidity $q_p$ of the parcels be uniformly distributed around the mean

specific humidity $q$. The limits of this distribution are $q_{\text{low}} := (1-a)q$ and $q_{\text{high}} := (1+a)q$, where $a < 1$ is a parameter to be chosen. These common assumptions about the clear sky sub grid fluctuations also come in handy in our case as they lead to a simple humidity distribution inside a subsequently generated cloud. Nevertheless they are a strong simplification of the fluctuations in the real atmosphere (cf. Gierens et al., 2007).

If we now let $T$ decrease, the saturation specific humidity with respect to ice, $q_s(T)$, is going to drop as well. We assume

here that cirrus forms by homogeneous nucleation only. Treatment of heterogeneous nucleation requires the treatment of solid aerosol as well, a complication that we want to avoid in the writing of this concept. Thus, cloud formation will not be initiated until the relative humidity with respect to ice, $RH_i$, in one of the air parcels passes the threshold for homogeneous freezing

specified by Kärcher and Lohmann (2002):

$$RH_{\mathrm{nuc}}(T) = 2.583 - \frac{T}{207.8 K} \tag{1}$$

$$q_{\mathrm{nuc}}(T) = RH_{\mathrm{nuc}}(T)\, q_s(T) \tag{2}$$

Other formulations for this threshold can be used as well. Let the time of the first nucleation event be $t_0$. From this time on, more and more air parcels within the grid box will exceed the nucleation threshold and begin to deposit their water vapour on ice crystals as long as $T$ keeps decreasing. Although the deposition of vapour on ice can be described in detail, e.g. taking into account ice crystal shape, accommodation coefficients of mass and heat, and so on (for details, see any textbook on cloud physics), we will assume here a basic formulation, namely that the vapour deposition is simply proportional to the current supersaturation (cf. Khvorostyanov and Sassen, 1998a, Eq. 26), i.e.

$$\frac{\mathrm{d}q_p}{\mathrm{d}t} = -\alpha(q_p(t) - q_s(T)). \tag{3}$$

In this simple formulation, all the microphysical details are embedded in the value of $\alpha$, which is simply the reciprocal value of the so-called phase relaxation time (Khvorostyanov and Sassen, 1998a):

$$\alpha = 4\pi\, D\, N\, \overline{r}, \tag{4}$$

where $D$ is the temperature and pressure dependent water vapour diffusion coefficient, $N$ is the number concentration of ice crystals and $\overline{r}$ is the radius that they would obtain if the complete excess vapour was transferred into $N$ spherical crystals. $\alpha$ is therefore a constant for each time step for a certain cloud, but different clouds can have different values of $\alpha$. Influences of changes in $\alpha$ will be discussed in section 4.

For each air parcel that made it to a cloudy state, $q_p$ will hence start decaying exponentially towards $q_s$ from the respective parcel's individual nucleation time $t_{\mathrm{nuc}} \geq t_0$ on. The parcel's specific ice water content $q_{i,p}$ will grow at the same rate and after a small period of time $\delta t$ we have the following conditions:

$$q_p(t + \delta t) = q_p(t) + \delta q_p \tag{5}$$

$$q_{i,p}(t + \delta t) = q_{i,p}(t) - \delta q_p \qquad \text{where} \tag{6}$$

$$\delta q_p = -\alpha(q_p(t) - q_s(T))\,\delta t \tag{7}$$

However, $q_s$ itself will keep dropping as long as $T$ keeps falling. It is well known (e.g. Khvorostyanov and Sassen, 1998a, b; Gierens, 2003) that an equilibrium relative supersaturation, $S_{\mathrm{eq}}$, will be reached after a while (the ~~so-called~~ phase relaxation time~~, which is the reciprocal value of $\alpha$~~). Supersaturation here refers to

$$s := \frac{q - q_s}{q_s} = \frac{q}{q_s} - 1 = RH_i - 1. \tag{8}$$

$S_{\mathrm{eq}}$ is given by $(\mathrm{d}s/\mathrm{d}t)_{\mathrm{eq}} = 0$, which can be translated to the $q$-space as follows:

$$0 = (\mathrm{d}s/\mathrm{d}t)_{\mathrm{eq}} = \frac{\mathrm{d}}{\mathrm{d}t}\left(\frac{q_p - q_s}{q_s}\right)_{\mathrm{eq}} \tag{9}$$

Performing the derivatives leads to the following condition

$$\frac{\mathrm{d}\ln q_p}{\mathrm{d}t} = \frac{\mathrm{d}\ln q_s}{\mathrm{d}t} \quad \text{in equilibrium,} \tag{10}$$

that is, the relative rate at which the specific humidity decreases by ice growth equals the relative rate by which the saturation specific humidity decreases by cooling of the air. From this condition, the equilibrium supersaturation can be determined as

$$S_{\mathrm{eq}} = \left( \frac{\alpha}{-\mathrm{d}\ln q_s/\mathrm{d}t} - 1 \right)^{-1}. \tag{11}$$

Details of the derivation may be found in the appendix.

Whether an equilibrium exists or not depends on the relative sizes of the "deposition rate" $\alpha$ and the relative decrease rate of the saturation specific humidity. If this ratio approaches unity (from above), the term in brackets approaches zero and then the equilibrium saturation ratio grows to infinity. Such a situation needs strong cooling in a cloud with low crystal number density. Fortunately, strong cooling generally induces large crystal numbers which in turn lead to stronger consumption of the available supersaturation (Kärcher and Lohmann, 2002). This mechanism usually guarantees an equilibrium with a few percent supersaturation. Therefore we assume that an equilibrium always exists at $S_{\mathrm{eq}}$. Note that this equilibrium "supersaturation" becomes negative if the saturation specific humidity increases, i.e. if the cloud experiences warming. An example of the cooling-cloud formation-phase relaxation process is presented in Figure 1.

As the cloudy air parcels had their individual nucleation events at different times, their current supersaturation is spread over a certain range of values with a distribution that needs to be determined. In principle this can be achieved using the box model and making a histogram of the supersaturation values. For more complicated initial conditions than assumed here this might actually be necessary, but since we start from a uniform distribution of $q_p$ in the clear grid box (which is equivalent to a uniform distribution of the relative humidity at constant $T$ throughout the box, as assumed), one can derive an analytical expression for the supersaturation distribution in the cloudy air parcels, $f_{S_p}(s)$. At the moment of nucleation, the air parcel has a supersaturation $S_{\mathrm{nuc}}$; thereafter the supersaturation will approach the mentioned equilibrium value, $S_{\mathrm{eq}}$. As temperature changes little during a small time step, the change in $S_{\mathrm{eq}}$ is generally negligible and the corresponding change in $S_{\mathrm{nuc}}$ is small, cf. Fig. 1. We neglect these changes for simplicity. We rather assume that for a little while after nucleation the supersaturation, $s$, in a single air parcel evolves as follows:

$$s(t, t_{\mathrm{nuc}}) = S_{\mathrm{eq}} + (S_{\mathrm{nuc}} - S_{\mathrm{eq}})\, e^{-\alpha\,(t-t_{\mathrm{nuc}})}, \quad \text{for} \quad t > t_{\mathrm{nuc}}. \tag{12}$$

This means, the growing cloud always consists of young, highly supersaturated parts and old parts with a supersaturation close to the current $S_{\mathrm{eq}}$. After some time has passed, the corresponding humidity distribution within the cloud appears to be of hyperbolic shape with high probability densities close to $S_{\mathrm{eq}}$ and lower densities at higher supersaturation. In order to derive $f_{S_p}(s)$, we start from the corresponding distribution of individual nucleation times, which is approximately a uniform one due to the assumption of an initially uniform distribution of $q_p$ and a constant cooling rate during one time step. Let, as before, nucleation start at a time $t_0$ and let us consider the situation at a later time $t_1$, which is either within the same time step or the end of the current time step. Then the uniform distribution of nucleation times is

$$f_{t_{\mathrm{nuc}}}(t) = (t_1 - t_0)^{-1} \quad \text{for} \quad t \in [t_0, t_1]. \tag{13}$$

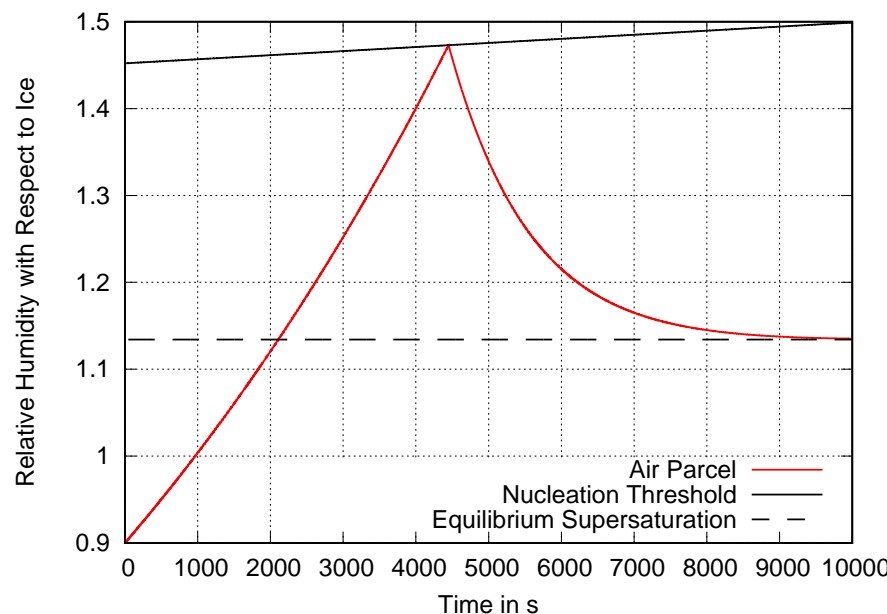

**Figure 1.** Example of the cooling-cloud formation-phase relaxation process. The red curve shows the supersaturation of a single air parcel as it increases over time, until it reaches the nucleation threshold (solid black). From there on the supersaturation is consumed by depositional growth of ice crystals, which lowers the supersaturation down to an equilibrium value (dashed black) a few percent above saturation.

We now use the transformation law for probability densities to derive $f_{S_p}(s)$ from $f_{t_{\mathrm{nuc}}}(t)$. Using Eq. 12 gives the relation between $S_p$ and $t_{\mathrm{nuc}}$:

$$\ln\left(\frac{s(t,t_{\mathrm{nuc}}) - S_{\mathrm{eq}}}{S_{\mathrm{nuc}} - S_{\mathrm{eq}}}\right) = -\alpha(t - t_{\mathrm{nuc}}) \tag{14}$$

from which we calculate the derivative

$$\frac{\mathrm{d}t_{\mathrm{nuc}}}{\mathrm{d}s} = \frac{1}{\alpha(s(t,t_{\mathrm{nuc}}) - S_{\mathrm{eq}})}. \tag{15}$$

With this, the transformation law reads

$$f_{S_p}(s) = f_{t_{\mathrm{nuc}}}[t(s)]\left|\frac{\mathrm{d}t_{\mathrm{nuc}}}{\mathrm{d}s}\right| \tag{16}$$

and the desired result is

$$f_{S_p}(s) = \frac{1}{\alpha(t_1 - t_0)(s - S_{eq})} \quad \text{for} \quad s \in [S_0, S_1], \tag{17}$$

where

$$S_0 = S_{\mathrm{eq}} + (S_{\mathrm{nuc}} - S_{\mathrm{eq}})\,e^{-\alpha(t_1 - t_0)} \quad \text{and} \quad S_1 = S_{\mathrm{nuc}}. \tag{18}$$

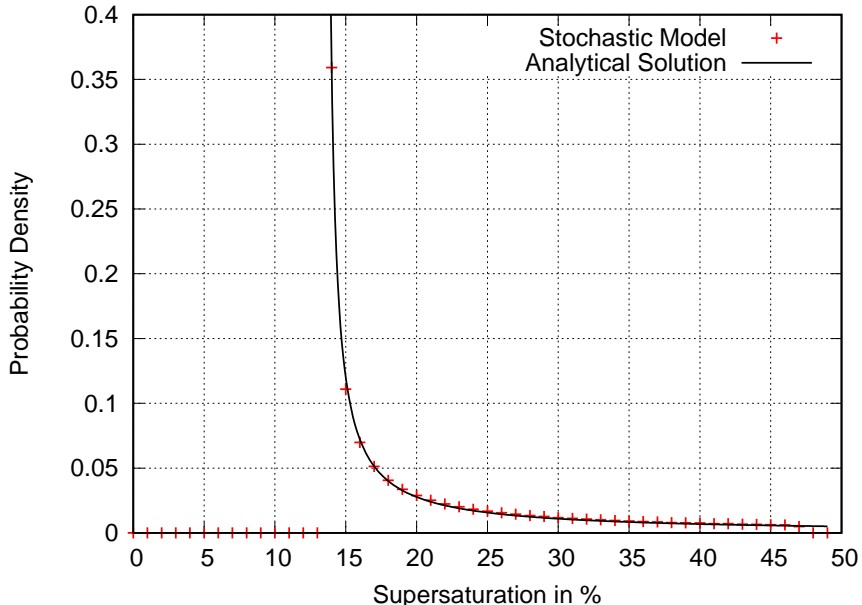

**Figure 2.** Hyperbolic distribution of supersaturation in the cloudy part of a grid box, analytical solution (solid line) and numerical result.

Fig. 2 shows that, despite the assumptions made, this analytical solution closely captures the probability distribution generated by the model.

With $S_0$ and $S_1$ one can write the distribution of supersaturation within the cloudy air parcels in terms of supersaturation values alone:

$$170 \quad f_{S_p}(s) = \frac{1}{\ln\left(\frac{S_1 - S_{\mathrm{eq}}}{S_0 - S_{\mathrm{eq}}}\right)(s - S_{eq})} \quad \text{for} \quad s \in [S_0, S_1]. \tag{19}$$

Note that later, after nucleation has stopped, $S_1$ will no longer be $S_{\mathrm{nuc}}$, it will evolve according to Eq. 12 instead, with $S_1 = s(t_1, t_{\mathrm{nuc}})$, but the general expression for the hyperbolic distribution of $S_p$ will still be valid.

It is easy to see that for large times $t_1$ the distribution of in-cloud supersaturation approaches a delta distribution centred at the equilibrium supersaturation:

$$175 \quad \lim_{t_1 \to \infty} f_{S_p}(s) = \delta(s - S_{\mathrm{eq}}). \tag{20}$$

This end point is higher than saturation as long as there is cooling and this is an important difference to saturation adjustment.

### 2.2 New parameterisation

We propose a new, simple ice cloud scheme that overcomes saturation adjustment by directly modelling the decay of supersaturation once cloud formation has begun at temperatures below the supercooling limit for small droplets at about $-38°C$. For 180 this purpose again consider a single model grid box with temperature $T$, specific humidity $q$, cloud fraction $C$ and specific ice

water content $q_i$. Let further $\delta T$ be the change in $T$ when integrating the next model time step and again imagine the grid box to consist of an arbitrarily large number of air parcels.

### 2.2.1 Initial cloud formation

Let us first consider $\delta T < 0$ and $C = 0$, thus $q_i = 0$. In this case we again assume $q_p$ to be uniformly distributed between $q_{\text{low}}$ and $q_{\text{high}}$ across the grid box. As $T$ decreases, $q_s(T)$ and $q_{\text{nuc}}(T)$ drop as well and cloud formation is initiated, if $q_{\text{nuc}}(T)$ passes $q_p$ at one point within the grid box. According to the sub grid humidity fluctuations, this happens, if $q_{\text{nuc}} < q_{\text{high}}$ at the end of the current time step, or equivalently if

$$q^n > \frac{q_{\text{nuc}}^{n+1}}{1+a}, \tag{21}$$

where upper indices $n, n+1$ label time steps in the following. The duration of one time step is $\Delta t = t^{n+1} - t^n$.

Consistent with the assumption of uniform $q$-fluctuations, the time, $t_0$, of the first air parcel becoming cloudy and the cloud fraction at the end of the time step can both be determined by simple geometric (i.e. linear) considerations. Assuming that $q_{\text{nuc}}$ decreases linearly with time during a time step gives

$$t_0 = t^n + \Delta t \frac{q_{\text{nuc}}^n - q_{\text{high}}}{q_{\text{nuc}}^n - q_{\text{nuc}}^{n+1}}. \tag{22}$$

Similarly we calculate the cloud fraction for the next time step as the difference between $q_{\text{high}}$ and $q_{\text{nuc}}^{n+1}$ divided by the total spread in $q_p$:

$$C^{n+1} = \frac{q_{\text{high}} - q_{\text{nuc}}^{n+1}}{2a\,q^n}. \tag{23}$$

With $t_0$ and the humidity distribution $f_{S_p}$ from equation 17 inside the new cloud, we can calculate the mean relative supersaturation $S_{\text{cl}}$ inside the cloud at $t^{n+1}$:

$$S_{\text{cl}}^{n+1} \qquad = \int_{S_0}^{S_1} s\,f_{S_p}(s)\,\mathrm{d}s \tag{24}$$

$$= S_{\text{eq}} + (S_{\text{nuc}} - S_{\text{eq}})\frac{1-\exp(-\alpha(t^{n+1}-t_0))}{\alpha(t^{n+1}-t_0)} \tag{25}$$

$$q_{\text{cl}}^{n+1} \qquad = (S_{\text{cl}}^{n+1} + 1)\,q_s^{n+1} \tag{26}$$

We leave it open here at which moment within a time step the quantities $S_{\text{nuc}}$ and $S_{\text{eq}}$ are evaluated. The choice has not much effect, since these quantities vary little over one time step. In our simulations, we always use the values at the beginning of the time step. The two limits of Eq. 25 are $S_{\text{eq}}$ for large time differences $t^{n+1} - t_0$ and $S_{\text{nuc}}$ for very small time differences, as expected. Note that we introduce $q_{\text{cl}}$ as a new prognostic variable at this point. This appears to be necessary in order to overcome saturation adjustment entirely.

The mean specific humidity in the clear air right after cloud formation, $q_{\text{env}}^{n+1}$, can be determined with a geometric consideration as well. Before cloud formation, the total spread in specific humidity was $2a\,q_{\text{env}}^n$. After partial cloud formation this

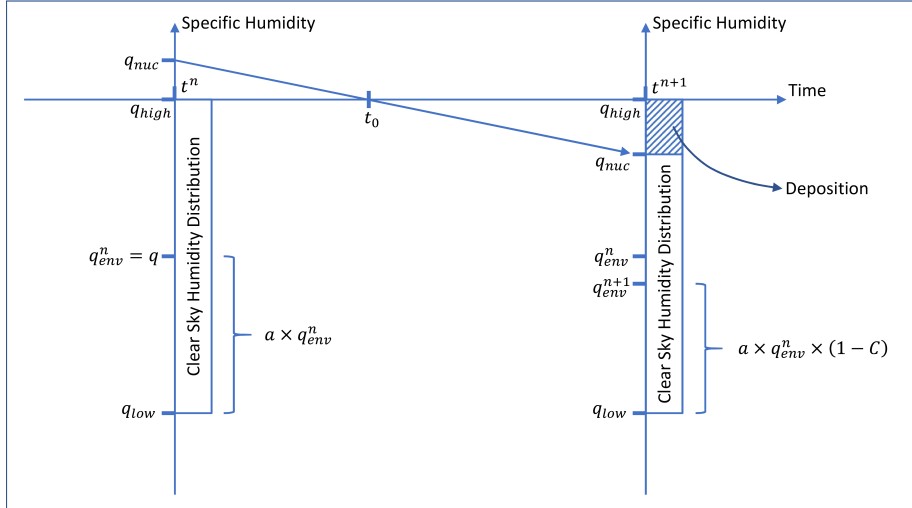

**Figure 3.** Schematic on the initiation of cloud formation in the parameterisation. At the time $t_0$, $q_{\mathrm{nuc}}$ propagates into the clear sky humidity distribution and thereby causes nucleation in the moistest part of the grid box. The parameterisation turns this part into a cloud and $q_{\mathrm{nuc}}$ becomes the new top end of the clear sky humidity distribution. The mean clear sky humidity $q_{\mathrm{env}}$ is adjusted accordingly.

spread is reduced by a factor $(1 - C^{n+1})$, such that

$$
\begin{aligned}
q_{\mathrm{env}}^{n+1} &= q_{\mathrm{low}} + a\, q_{\mathrm{env}}^{n}\,(1 - C^{n+1}) \\
&= q_{\mathrm{env}}^{n}\,(1 - a\, C^{n+1}).
\end{aligned}
\tag{27}
$$

Figure 3 shows schematically, how the clear sky humidity develops over time.

If cloud formation were so vigorous (or if the grid box is small with small $a$) that the cloud fraction became unity within the time step, then $q_{\mathrm{env}}^{n+1}$ would be undefined and the mean specific humidity in the grid box, $q$, would correspond to the mean supersaturation in the cloud. Otherwise, the mean specific humidity is the weighted mean of $q_{\mathrm{env}}^{n+1}$ and $q_{\mathrm{cl}}^{n+1}$:

$$
q^{n+1} = (1 - C^{n+1})\, q_{\mathrm{env}}^{n+1} + C^{n+1}\, q_{\mathrm{cl}}^{n+1}.
\tag{28}
$$

For the mean specific ice water content $q_i^{n+1}$, we know that all water vapour that leaves the gas phase is deposited as ice and thus

$$
q_i^{n+1} = q^n - q^{n+1}.
\tag{29}
$$

Note that these equations are valid at the end of the time step at which cloud formation commences. Equations for later time steps will be derived below.

### 2.2.2 Continual cloud growth

Let us now consider further time steps in which cloud formation continues and the cloud fraction increases. The general time step ranges from $t^{n+i}$ to $t^{n+i+1}$, $\quad i \geq 1$, but let us take $i = 1$ for simplicity of notation. We now use our new prognostic variable $q_{\mathrm{cl}}$ to recover the mean specific humidity in the clear sky part of the grid box by rearranging equation 28:

$$q_{\mathrm{env}}^{n+1} = \frac{q^{n+1} - C^{n+1} q_{\mathrm{cl}}^{n+1}}{1 - C^{n+1}} \tag{30}$$

It is necessary, to formulate all changes in terms of quantities at the two considered time steps, so that not too many quantities have to be kept in memory. Again, determining the new cloud fraction, $C^{n+2}$ is a simple geometric task and the result is

$$C^{n+2} = C^{n+1} + (q_{\mathrm{nuc}}^{n+1} - q_{\mathrm{nuc}}^{n+2}) \frac{1 - a\, C^{n+1}}{2a\, q_{\mathrm{env}}^{n+1}}. \tag{31}$$

With this, the new average specific humidity in the cloud free part, $q_{\mathrm{env}}^{n+2}$, can be computed in analogy to Eq. 27:

$$q_{\mathrm{env}}^{n+2} = q_{\mathrm{env}}^{n} (1 - a\, C^{n+2}), \tag{32}$$

but here it is necessary to replace $q_{\mathrm{env}}^{n}$ by its updated value from Eq. 27, giving

$$q_{\mathrm{env}}^{n+2} = q_{\mathrm{env}}^{n+1} \frac{1 - a\, C^{n+2}}{1 - a\, C^{n+1}}. \tag{33}$$

For the in-cloud humidity, we split the grown cloud into the old part with cloud fraction $C^{n+1}$ and the new part with cloud fraction $C^{n+2} - C^{n+1}$. In the new part, cloud formation is initiated as before according to eqs. 24 to 26, but now nucleation occurs right from the beginning of the time step, giving

$$S_{\mathrm{new}} \quad = S_{\mathrm{eq}} + (S_{\mathrm{nuc}} - S_{\mathrm{eq}}) \frac{1 - \exp(-\alpha\,\Delta t)}{\alpha\,\Delta t} \tag{34}$$

$$q_{\mathrm{new}} \qquad = (S_{\mathrm{new}} + 1) q_s^{n+2}. \tag{35}$$

The old cloud part on the other hand can be considered as an individual cloud that stopped growing at $t^{n+1}$. If a cloud stops growing, not only the lower boundary $S_0$ of the in-cloud distribution of supersaturation further approaches $S_{\mathrm{eq}}$ but the upper boundary $S_1$ starts to deviate from $S_{\mathrm{nuc}}$, as mentioned above. Both boundaries develop in a similar way, the lower one proceeds further as in Eq. 18, with the end time updated. The upper one proceeds in an analogous way, however, with $t_0$ replaced by $t^{n+1}$. In this case, we have

$$S_{\mathrm{old}} \qquad = \int_{S_p(t^{n+2}, t_0)}^{S_p(t^{n+2}, t^{n+1})} s\, f_{S_p}(s)\, \mathrm{d}s \tag{36}$$

$$= S_{\mathrm{eq}} + (S_{\mathrm{nuc}} - S_{\mathrm{eq}}) \frac{\left(\exp(\alpha t^{n+1}) - \exp(\alpha t_0)\right) \exp(-\alpha t^{n+2})}{\alpha(t^{n+1} - t_0)}. \tag{37}$$

In order to evaluate this, one would need to have $t_0$ still in memory. Instead, we recognize that this is a simple exponential decay towards $S_{\mathrm{eq}}$, as we would expect if $q_p$ decays exponentially in all the air parcels. The specific humidity $q_{\mathrm{old}}$ in the old cloud part can thus simply be calculated as

$$q_{\mathrm{old}} = q_{\mathrm{cl}}^{n+1} - \alpha\,\Delta t(q_{\mathrm{cl}}^{n+1} - q_s^{n+1}). \tag{38}$$

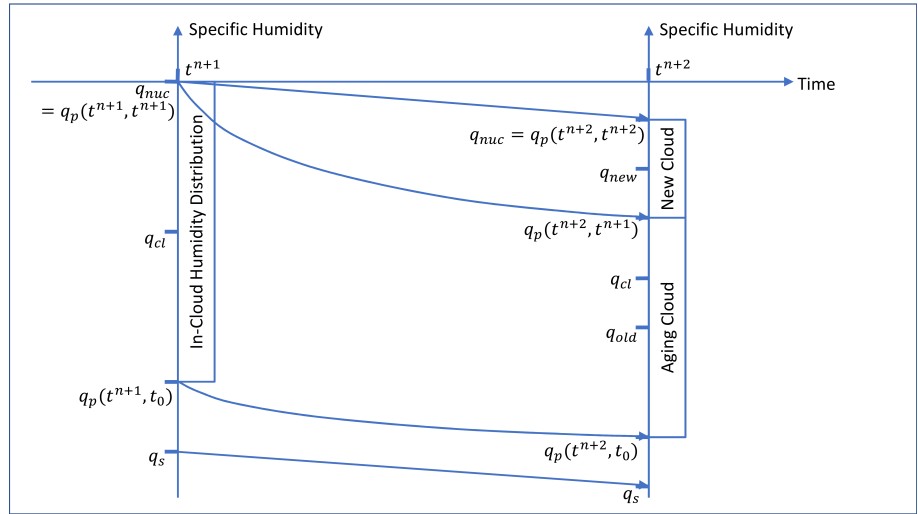

**Figure 4.** Schematic on the evolution of the in-cloud humidity during ongoing cloud formation. The boundaries of the humidity distribution in the already existing cloud decay exponentially towards $q_s$, while the steady decrease of $q_{\mathrm{nuc}}$ continuously causes nucleation in further parts of the clear sky portion of the grid box. These new cloud parts begin their own decay towards $q_s$ at their individual nucleation times resulting in a steady new humidity distribution in the grown cloud with the new $q_{\mathrm{nuc}}$ as its top end.

Putting the cloud together again, we can derive

$$q_{\mathrm{cl}}^{n+2} = \frac{C^{n+1}}{C^{n+2}} \, q_{\mathrm{old}} + \frac{C^{n+2} - C^{n+1}}{C^{n+2}} \, q_{\mathrm{new}}. \tag{39}$$

Figure 4 shows schematically, how the in-cloud humidity develops over time.

The mean specific humidity in the grid box, $q^{n+2}$ is as before the weighted mean of $q_{\mathrm{cl}}^{n+2}$ and $q_{\mathrm{env}}^{n+2}$, and the new specific ice water content is given by

255 $$q_i^{n+2} = q_i^{n+1} + q^{n+1} - q^{n+2}. \tag{40}$$

### 2.2.3 Continual cloud growth to full coverage

If full cloud coverage is reached within a time step, it is necessary to record the point in time, $t_1$, when this happens. This can again be done by a geometric consideration. Let the current time step start at $t^m$. The current clear-sky specific humidity $q_{\mathrm{env}}^m$ can again be calculated according to equation 30. With this, $t_1$ becomes

$$t_1 = t^m + \Delta t \frac{2(q_{\mathrm{nuc}}^m - q_{\mathrm{env}}^m)}{q_{\mathrm{nuc}}^m - q_{\mathrm{nuc}}^{m+1}}. \tag{41}$$

Now one computes the average supersaturation in the new cloud part at $t^{m+1}$ in two steps. The first step ranges from $t^m$ to $t_1$, for which the upper boundary of $f_{S_p}$ is still the current nucleation threshold. This gives a preliminary supersaturation, $S^*_{\mathrm{new}}$

$$S^*_{\mathrm{new}} = S_{\mathrm{eq}} + (S_{\mathrm{nuc}} - S_{\mathrm{eq}}) \frac{1 - \exp(-\alpha(t_1 - t^m))}{\alpha(t_1 - t^m)} \tag{42}$$

$$q^*_{\mathrm{new}} = (S^*_{\mathrm{new}} + 1)\, q^m_s. \tag{43}$$

The equilibrium and nucleation supersaturations have to be taken at the conditions of the current time step. After nucleation has ceased, that is, in the second step from $t_1$ to $t^{m+1}$, the new cloud part can be treated like the old one:

$$q_{\mathrm{new}} = q^*_{\mathrm{new}} - \alpha(t^{m+1} - t_1)(q^*_{\mathrm{new}} - q^m_s) \tag{44}$$

The specific humidity in the old cloud part can be calculated as before, such that we have for $q^{m+1}$:

$$q^{m+1} = C^m\, q_{\mathrm{old}} + (1 - C^m)\, q_{\mathrm{new}} \tag{45}$$

$C^{m+1}$ can then be set to one and $q_i^{m+1}$ can be calculated as before.

### 2.2.4 Completely covered sky and cooling

Let us now consider a time step $t^{m+i}$ to $t^{m+i+1}$ after full coverage has been reached, where we again take $i = 1$ for simplicity of notation. In this simple case we only let $q$ decay further towards $q_s$, while $q_s$ itself decreases

$$q^{m+2} = q^{m+1} - \alpha\,\Delta t(q^{m+1} - q_s^{m+1}) \tag{46}$$

and calculate $q_i^{m+2}$ analogously to equation 40.

### 2.2.5 Warming a cloud

Now consider a time step $t^k$ to $t^{k+1}$, during which there is a cloud inside the grid box and $\delta T > 0$. If the grid box mean temperature keeps rising over a longer period of time, the ice crystals inside the cloud are expected to return to the gas phase and eventually vanish. During this process, it is hard to determine a threshold for $q_{i,p}$ below which a specific air parcel can be

considered as no longer cloudy. Since we also neglect cloud edge erosion in this concept, we simply do not let any air parcel leave the cloud and keep $C$ constant until all ice has sublimated.

For the in-cloud humidity, we again have the conditions of a no longer growing cloud, i.e. $S_{\mathrm{cl}}$ decays exponentially towards $S_{\mathrm{eq}}$. Therefore, $q_{cl}$ evolves analogously to equation 46:

$$q_{\mathrm{cl}}^{k+1} = q_{\mathrm{cl}}^k - \alpha\,\Delta t(q_{\mathrm{cl}}^k - q_s^k) \tag{47}$$

$q^{k+1}$ and $q_i^{k+1}$ can then be calculated via

$$q^{k+1} = q^k + \delta q \tag{48}$$

$$q_i^{k+1} = q_i^k - \delta q, \qquad \text{where} \tag{49}$$

$$\delta q = C(q_{\mathrm{cl}}^{k+1} - q_{\mathrm{cl}}^k). \tag{50}$$

**Table 1.** Overview of the tests

| Up/Down-draught | $w(t)$ (cm/s) | $\alpha$ (1/s) | $a$ | $\Delta t$ (s) |
|---|---|---|---|---|
| Scenario 1 (Fig. 6) | const (2) | $3 \cdot 10^{-4}$ | 0.25 | 60 |
| Scenario 2 (Fig. 7) | cos | $3 \cdot 10^{-4}$ | 0.25 | 60 |
| Scenario 1 (Fig. 8) | const (2) | $2.8 \cdot 10^{-3}$ | 0.25 | 60 |
| Scenario 2 (Fig. 9) | const (2) | $2.8 \cdot 10^{-3}$ | 0.25 | 60 |
| Scenario 1 (Fig. 10) | const (2) | $3 \cdot 10^{-4}$ | 0.10 | 60 |
| Scenario 2 (Fig. 11) | cos | $3 \cdot 10^{-4}$ | 0.10 | 60 |
| Scenario 1 (Fig. 12) | const (10) | $3 \cdot 10^{-4}$ | 0.25 | 60 |
| Scenario 1 (Fig. 13) | const (2) | $3 \cdot 10^{-4}$ | 0.25 | 600 |
| Scenario 2 (Fig. 14) | cos | $3 \cdot 10^{-4}$ | 0.25 | 600 |

If $\delta q > q_i^k$, the cloud is about to vanish in the course of the current time step. In this case, we instead simply set

$$q^{k+1} = q^k + q_i^k \tag{51}$$

$$q_i^{k+1} = 0 \tag{52}$$

$$C^{k+1} = 0. \tag{53}$$

## 3  Results

The new parameterisation has been tested against the stochastic box model and against a parameterisation with saturation adjustment in various scenarios of cloud formation and dissolution, which are essentially distinguished by the temporal behaviour of the vertical velocity, $w(t)$, or equivalently, of the adiabatic cooling and warming over time. Two such scenarios are considered, one with a constant updraught, and one where $w(t)$ follows a cosine function with two different amplitudes (cf. fig. 5, left panel). These two scenarios are additionally combined with variations of the other parameters, i.e., $\alpha$, $a$, and $\Delta t$, in order to test the sensitivity of the parameterisation to variations of these parameters. Table 1 gives an overview of the performed simulations. The right panel of fig. 5 shows exemplarily how the cloud fraction evolves in the two wind scenarios with the parameter setting of the first two simulations.

All tests are illustrated with pairs of figures, where each left panel shows the temporal evolution of the grid-mean relative humidity, which is an output-variable of NWP models. Each right panel shows the temporal evolution of the in-cloud relative humidity, which is (proportional to) the new prognostic variable in our concept. All simulations start from an initial grid box mean temperature of 235 K so that the temperature at the time of first nucleation will always be below the threshold for homogeneous freezing.

The first simulation uses a constant uplift speed of $2\,\mathrm{cm\,s^{-1}}$ and ice growth with $\alpha = 3 \cdot 10^{-4}\,\mathrm{s^{-1}}$. The value of $\alpha$ is chosen by taking into account the results of Khvorostyanov and Sassen (1998b). According to their table A1, this value would be appropriate, for example, if about 100 ice crystals with a mean radius of $10\,\mu\mathrm{m}$ were present per litre of air. We set $a = 0.25$,

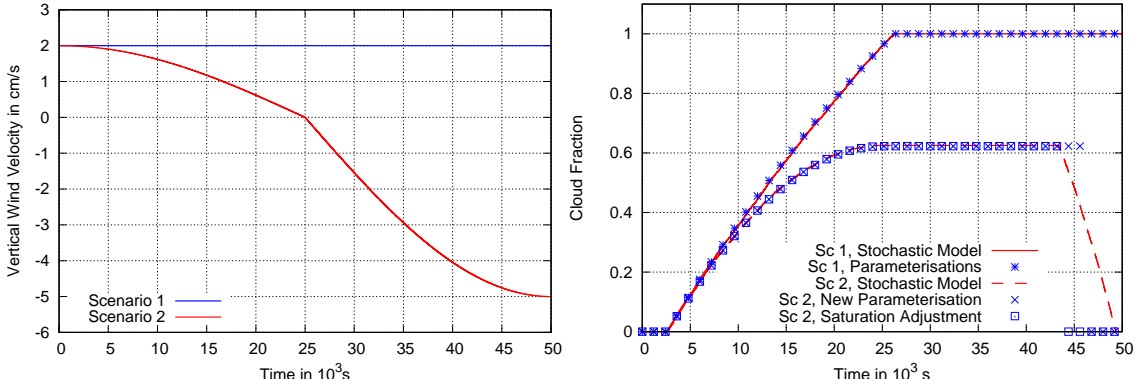

**Figure 5.** Left Panel: Visualisation of the considered vertical wind speed scenarios. Scenario 1 is characterized by a constant updraught, whereas the vertical wind speed in scenario 2 follows half a period of a cosine function in time with an increase in amplitude half way through the simulation from $2\,\mathrm{cm\,s}^{-1}$ to $5\,\mathrm{cm\,s}^{-1}$. Right Panel: The corresponding evolutions of cloud fraction in the stochastic model and the parameterisations for $\alpha = 3\cdot10^{-4}\,\mathrm{s}^{-1}$ and $a = 0.25$. As can be seen, the parameterisations both capture the evolution of cloud fraction precisely. Only the process of cloud dissolution is not modelled accurately, as the cloud fraction is simply set to zero, once all ice has sublimated. Note that the sublimation process is slower in the new parameterisation compared to the one using saturation adjustment which increases the cloud life time.

such that nucleation is initiated once $q > \frac{q_{\mathrm{nuc}}}{1+0.25} = 0.8\,q_{\mathrm{nuc}}$ which is the current standard in the integrated forecast system. The time step is $60\,\mathrm{s}$. Figure 6 illustrates two mechanisms how saturation adjustment leads to an underestimation of supersaturation in cloudy situations. Right from the initiation of cloud formation in a grid box, saturation adjustment drives the mean relative humidity over ice down to 100%. As soon as full cloud coverage is reached, the mean relative humidity is and stays at 100%. To the contrary, the stochastic box model (which is our benchmark) shows that the humidity still increases after cloud initiation,

reaches a maximum and then approaches the equilibrium supersaturation a few percent above 100%. The new parameterisation closely follows this behaviour and thus represents reality better than saturation adjustment. The two reasons for underestimation of supersaturation in the current parameterisations with saturation adjustment are: 1) decrease of supersaturation right at cloud initiation and 2) ignorance of the equilibrium supersaturation later. This becomes particularly evident in the right panel of Fig. 6 where the in-cloud relative humidity is plotted for this case. With saturation adjustment, there is no in-cloud supersaturation,

but in reality there is at least as long as cooling proceeds.

The second test is a situation where the cooling rate follows a cosine function that changes amplitude half way through the simulation, that is, the simulation starts with cooling through an updraught of $2\,\mathrm{cm\,s}^{-1}$ which gets weaker over time and eventually reverses into warming and ice sublimation in a downdraught of finally $5\,\mathrm{cm\,s}^{-1}$. The temporal evolution of relative humidity (grid-box average and in-cloud average) is shown in Fig. 7. Clearly, saturation adjustment also shows a

325 deviating behaviour under the warming conditions in the second half of the simulation. In the stochastic box model and the new parameterisation, the in-cloud RHi approaches the equilibrium humidity, which is now below saturation, while the cloud again is assumed to be exactly saturated in the model with saturation adjustment. Sublimation is, just as deposition, only effective if

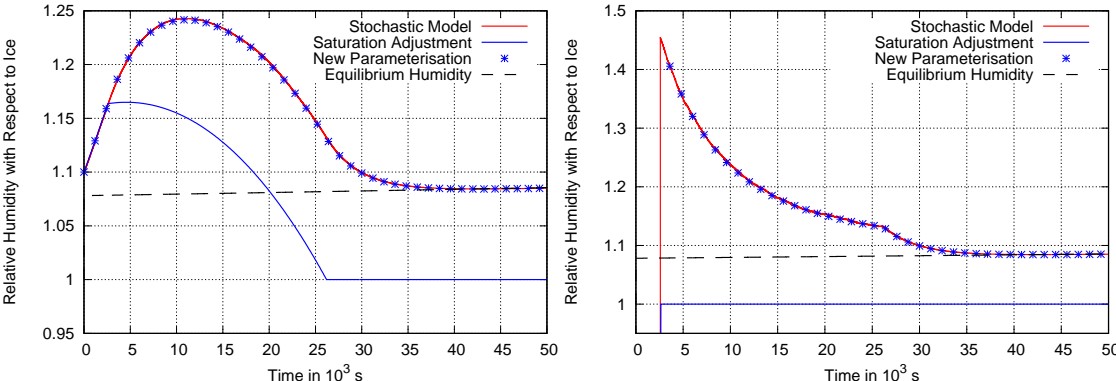

**Figure 6.** Scenario 1: Left panel: Grid box mean relative humidity (with respect to ice) as a function of time in a grid box with $a = 0.25$ that undergoes cooling in a constant updraught of $2 \, \mathrm{cm \, s^{-1}}$ and cloud formation with $\alpha = 3 \cdot 10^{-4} \mathrm{s^{-1}}$. Shown is the behaviour of the stochastic box model that treats one grid box as an ensemble of $10^4$ air parcels with an initial distribution of specific humidity (red); this model is assumed to best represent reality and serves as a benchmark for the parameterisations. The new parameterisation (blue stars) follows the stochastic model closely, while the parameterisation that uses saturation adjustment does not, which leads to an underestimation of the mean relative humidity and thus of supersaturation. The new parameterisation approaches the equilibrium supersaturation (black dashed), while saturation adjustment assumes exactly saturation inside the cloud. Right panel: The corresponding mean in-cloud relative humidity for the stochastic model, the new parameterisation, and the parameterisation with saturation adjustment. One can clearly see, how the decay of humidity in the stochastic box model changes into a pure exponential, once the cloud fraction reaches unity and no new, highly supersaturated cloud parts join the cloud anymore (cf. fig. 5, right panel).

the relative humidity deviates from saturation and it is not an instantaneous process in reality. Therefore, subsaturation inside the cloud can be considered the more realistic state and thus the new parameterisation shows an improvement to saturation adjustment also under warming conditions.

Next, cases with higher ice formation rate $\alpha = 2.8 \cdot 10^{-3} \mathrm{s^{-1}}$ are considered, which corresponds to clouds with 1000 ice crystals of 10 $\mu$m radius per litre. The corresponding phase relaxation time is only 6 min. Figs. 8 and 9 show that quite small differences between the old and the new parameterisation and the stochastic box model remain if the in-cloud supersaturation is consumed quickly, as expected. Of course, right after cloud formation, there is considerable supersaturation in the cloud which is not at all represented by the old parameterisation (Figs. 8 and 9, right panels); however, as the cloud fraction is initially small, this does not cause a large difference in the mean specific humidity (cf. the corresponding left panels). Thus, saturation adjustment leads to only a small underestimation of supersaturation in cases of clouds with quick vapour deposition on ice.

As the results of Gierens et al. (2007) suggest, the actual spread in $q$ across a small model grid box (i.e. higher resolution) is probably smaller than $\pm 25\%$. Figs. 10 and 11 show that the deviation of the parameterisation with saturation adjustment from the stochastic box model increases in a case with a narrower initial distribution of $q$ (here $\pm 10\%$). The maximum deviation in the constant updraught simulation increases from less than 15 percentage points in figure 6 to about 20 percentage points in figure 10 and the maximum deviation in the modulated updraught simulation increases from less than 10 percentage points

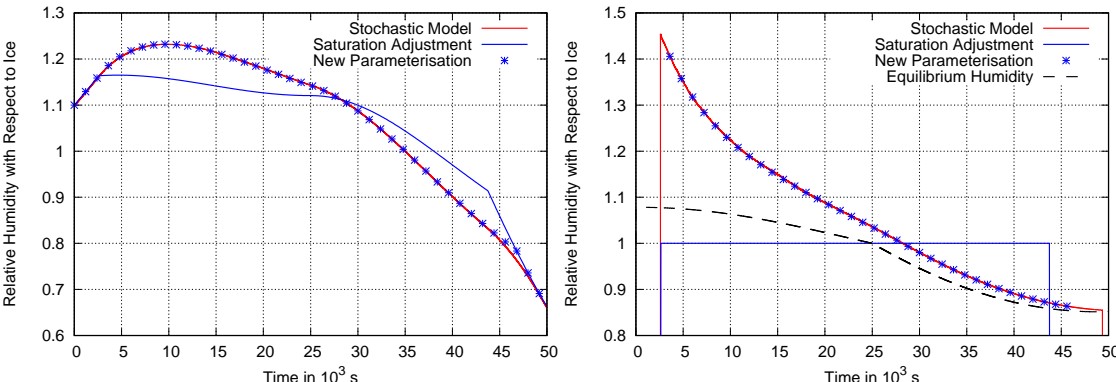

**Figure 7.** Scenario 2: Left panel: Grid box mean relative humidity (with respect to ice) as a function of time in an updraught of initially $2\,\mathrm{cm\,s^{-1}}$ that weakens over time turning into an accelerating downdraught of eventually $5\,\mathrm{cm\,s^{-1}}$ (cf. fig. 5). Other simulation parameters are $\alpha = 3 \cdot 10^{-4}\,\mathrm{s^{-1}}$ and $a = 0.25$. The behaviour of the benchmark box model is shown in red. The new parameterisation (blue stars) follows closely, while the parameterisation that uses saturation adjustment does not, which leads to an underestimation of relative humidity during the cooling period and an overestimation during the warming period. Right panel: The corresponding mean in-cloud relative humidity for the stochastic model, the new parameterisation, and the parameterisation with saturation adjustment. The box model and the new parameterisation approach the equilibrium supersaturation (black dashed), while saturation adjustment assumes exactly saturation inside the cloud.

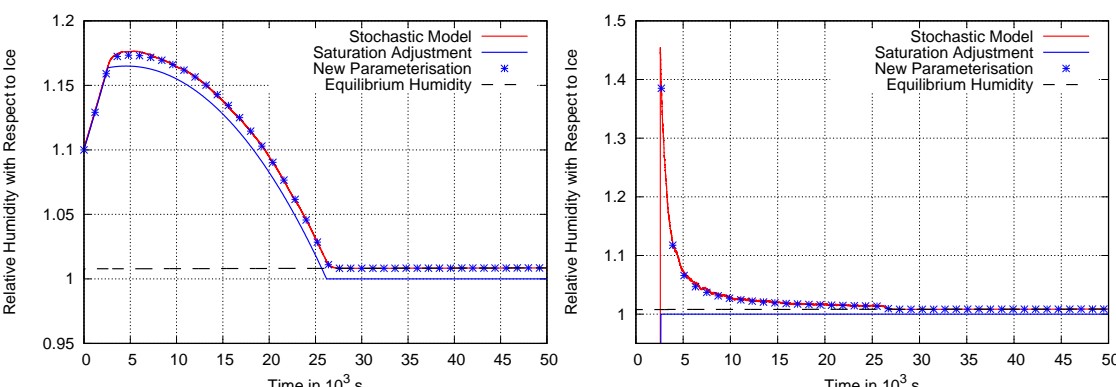

**Figure 8.** Scenario 1: Left panel: Grid box mean relative humidity (with respect to ice) as a function of time in a constant updraught of $2\,\mathrm{cm\,s^{-1}}$ with $a = 0.25$. The assumed deposition rate is high, $\alpha = 2.8 \cdot 10^{-3}\,\mathrm{s^{-1}}$, which corresponds to a cloud with 1000 crystals of $10\,\mu\mathrm{m}$ radius per litre. The behaviour of the benchmark box model is shown in red. The new parameterisation (blue stars) follows closely, while the parameterisation that uses saturation adjustment does not. This leads to only a slight underestimation of relative humidity due to the large deposition rate. Right panel: The corresponding mean in-cloud relative humidity for the stochastic model, the new parameterisation, and the parameterisation with saturation adjustment. The box model and the new parameterisation approach the equilibrium supersaturation (black dashed), while saturation adjustment assumes exactly saturation inside the cloud.

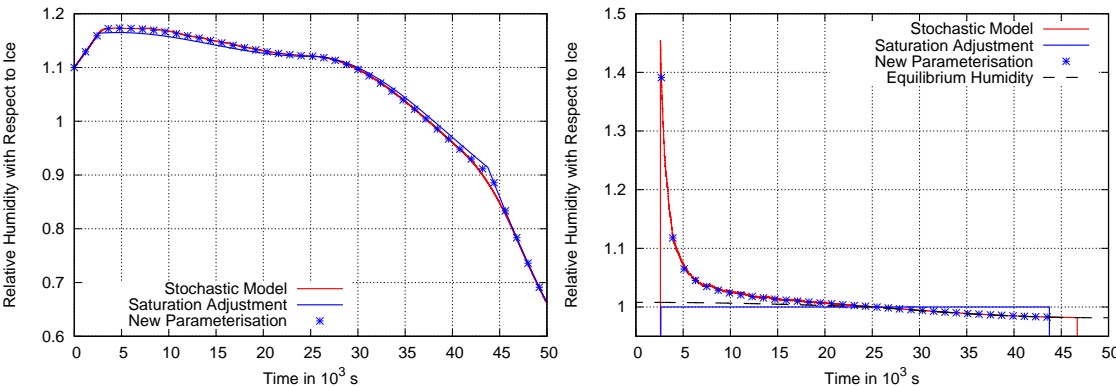

**Figure 9.** Scenario 2: Left panel: Grid box mean relative humidity (with respect to ice) as a function of time with $a = 0.25$ in an updraught of initially $2\,\mathrm{cm\,s^{-1}}$ that weakens over time turning into an accelerating downdraught of eventually $5\,\mathrm{cm\,s^{-1}}$ (cf. fig. 5). The assumed deposition rate is high, $\alpha = 2.8 \cdot 10^{-3}\,\mathrm{s^{-1}}$, which corresponds to a cloud with 1000 crystals of 10 $\mu$m radius per litre. The behaviour of the benchmark box model is shown in red which is closely followed by the new parameterisation (blue stars). The parameterisation that uses saturation adjustment only slightly underestimates relative humidity during the cooling period and slightly overestimates it during the warming period due to the large deposition rate. Right panel: The corresponding mean in-cloud relative humidity for the stochastic model, the new parameterisation, and the parameterisation with saturation adjustment. The box model and the new parameterisation approach the equilibrium supersaturation (black dashed), while saturation adjustment assumes exactly saturation inside the cloud.

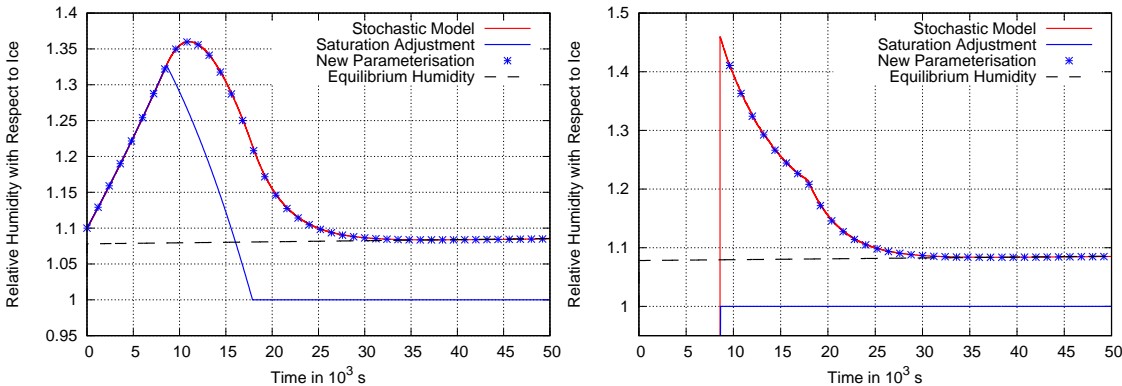

**Figure 10.** Scenario 1: Left panel: Grid box mean relative humidity (with respect to ice) as a function of time in a constant updraught of $2\,\mathrm{cm\,s^{-1}}$. The assumed deposition rate is $\alpha = 3 \cdot 10^{-4}\,\mathrm{s^{-1}}$. The initial humidity distribution is narrow with $a = 0.1$. The behaviour of the benchmark box model is shown in red. The new parameterisation (blue stars) follows closely, while the parameterisation that uses saturation adjustment does not, which leads to a larger underestimation of relative humidity compared to the reference simulation in fig. 6. Right panel: The corresponding mean in-cloud relative humidity for the stochastic model, the new parameterisation, and the parameterisation with saturation adjustment. The box model and the new parameterisation approach the equilibrium supersaturation (black dashed), while saturation adjustment assumes exactly saturation inside the cloud.

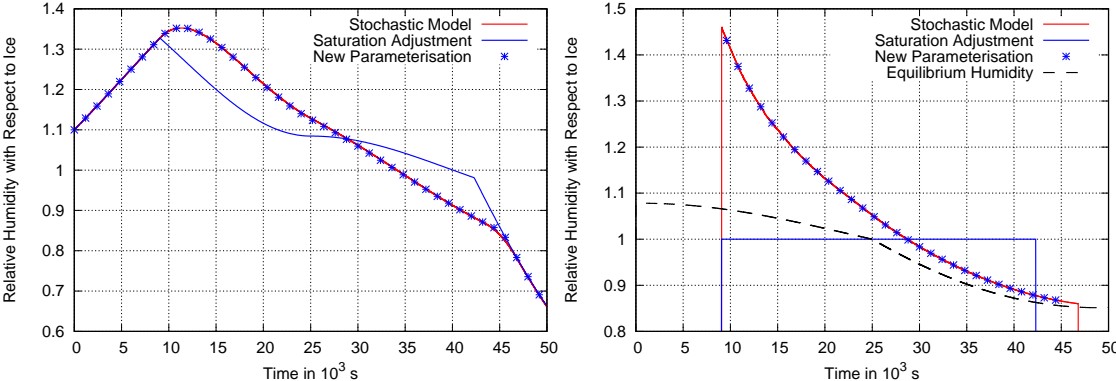

**Figure 11.** Scenario 2: Left panel: Grid box mean relative humidity (with respect to ice) as a function of time in an updraught of initially $2\,\mathrm{cm\,s^{-1}}$ that weakens over time turning into an accelerating downdraught of eventually $5\,\mathrm{cm\,s^{-1}}$ (cf. fig. 5). The assumed deposition rate is $\alpha = 3\cdot 10^{-4}\,\mathrm{s^{-1}}$. The initial humidity distribution is narrow with $a = 0.1$. The behaviour of the benchmark box model is shown in red which is closely followed by the new parameterisation (blue stars). The parameterisation that uses saturation adjustment underestimates relative humidity during the cooling period and overestimates it during the warming period more strongly due to the narrower clear sky humidity distribution. Right panel: The corresponding mean in-cloud relative humidity for the stochastic model, the new parameterisation, and the parameterisation with saturation adjustment. The box model and the new parameterisation approach the equilibrium supersaturation (black dashed), while saturation adjustment assumes exactly saturation inside the cloud.

(fig. 7) to more than 10 points (fig. 11). The reason is that the cloud fraction increases faster if the spread in $q$ is smaller. Thus, the initially high supersaturation inside the cloud in the stochastic box model gains more weight in the grid box mean relative humidity, whereas saturation adjustment drives the grid box mean relative humidity even faster back to $100\%$.

The equilibrium supersaturation rises quickly with increasing cooling rates. Figure 12 shows that it can even rise beyond the nucleation threshold in a rather extreme case of $10\,\mathrm{cm\,s^{-1}}$ constant updraught. In such a situation, the deposition of water vapour on the ice crystals is not fast enough to make the supersaturation inside the cloud decay; it can only slow down the further increase. Here, saturation adjustment does not only lead to wrong values of the grid box mean relative humidity but even to a wrong trend. Anyway, this simulation soon becomes unrealistic, as the high supersaturation would let the ice crystals grow quickly in size and number. Thereby, the available surface for vapour deposition would increase and the value of $\alpha$ would grow larger and larger. Hence, the equilibrium supersaturation would soon decrease again and the state of steadily increasing supersaturation could no longer be maintained.

A time step of one minute is rather short in numerical weather prediction. We therefore also test the sensitivity of the new parameterisation to the length of the time step by repeating the two experiments with a time step of ten minutes. The results are shown in figs. 13 and 14. Comparing these figures to figs. 6 and 7, one can clearly see that the sensitivity is low. The new parameterisation still precisely captures the onset of cloud formation and also later on closely follows the behaviour of the stochastic box model. The slight deviation by the end of the modulated cooling simulation (fig. 14, left panel) is not due to an error in the specific humidity, as the three models show identical final values in a plot of specific humidity (not shown). This

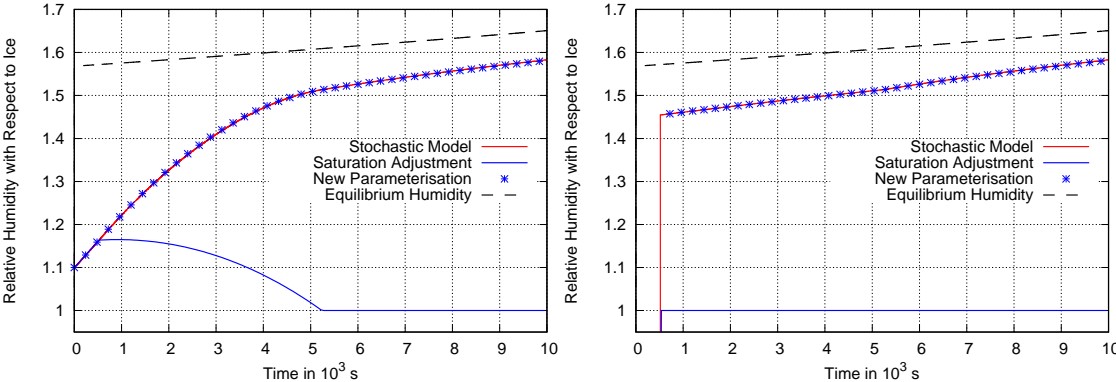

**Figure 12.** Scenario 1: Left panel: Grid box mean relative humidity (with respect to ice) as a function of time in a fast constant updraught of $10\,\mathrm{cm\,s^{-1}}$. Other simulation parameters are $\alpha = 3 \cdot 10^{-4}\,\mathrm{s^{-1}}$ and $a = 0.25$. The behaviour of the benchmark box model is shown in red. It increases steadily and is closely followed by the new parameterisation (blue stars), while the parameterisation that uses saturation adjustment does not even reproduce the same trend. The new parameterisation is hence in principle capable of simulating large supersaturations beyond the threshold for homogeneous nucleation occasionally found in the real atmosphere. However, the box model result quickly becomes unrealistic, as the increasing supersaturation would cause $\alpha$ to grow due to a rapid increase in crystal number density. Right panel: The corresponding mean in-cloud relative humidity for the stochastic model, the new parameterisation, and the parameterisation with saturation adjustment. The box model and the new parameterisation approach the equilibrium supersaturation (black dashed), while saturation adjustment assumes exactly saturation inside the cloud.

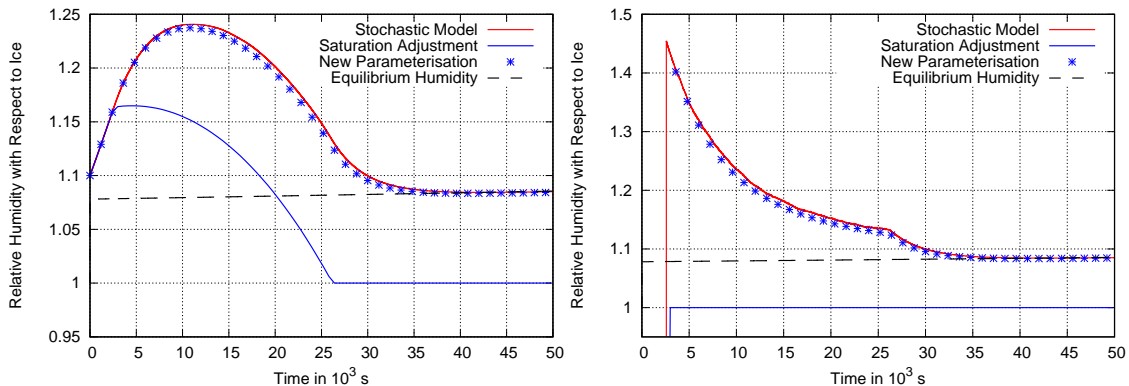

**Figure 13.** Scenario 1: Left panel: Grid box mean relative humidity (with respect to ice) as a function of time in a constant updraught of $2\,\mathrm{cm\,s^{-1}}$. Other simulation parameters are $\alpha = 3 \cdot 10^{-4}\,\mathrm{s^{-1}}$ and $a = 0.25$. The time step of the parameterisations is increased to 10 min. The behaviour of the benchmark box model is shown in red. The new parameterisation (blue stars) still follows quite closely, while the parameterisation that uses saturation adjustment does not, which leads to an underestimation of relative humidity. Right panel: The corresponding mean in-cloud relative humidity for the stochastic model, the new parameterisation, and the parameterisation with saturation adjustment. The box model and the new parameterisation approach the equilibrium supersaturation (black dashed), while saturation adjustment assumes exactly saturation inside the cloud.

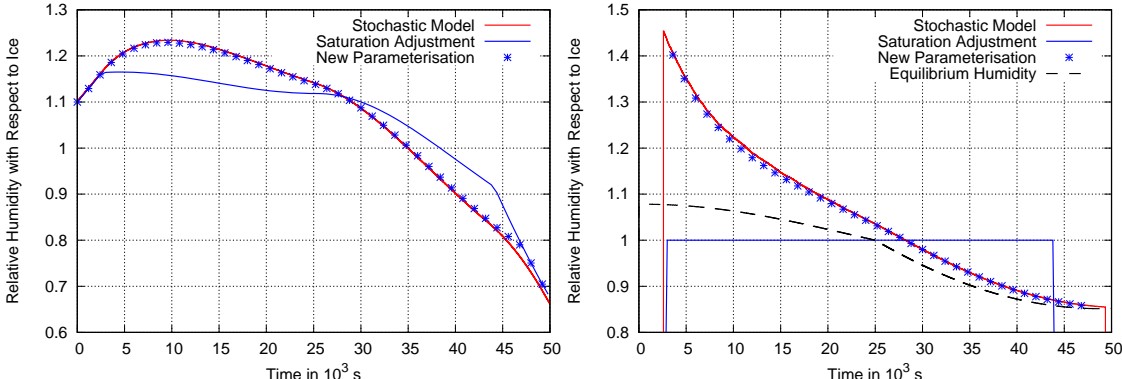

**Figure 14.** Scenario 2: Left panel: Grid box mean relative humidity (with respect to ice) as a function of time in an updraught of initially $2\,\mathrm{cm\,s^{-1}}$ that weakens over time turning into an accelerating downdraught of eventually $5\,\mathrm{cm\,s^{-1}}$ (cf. fig. 5). Other simulation parameters are $\alpha = 3\cdot10^{-4}\,\mathrm{s^{-1}}$ and $a = 0.25$. The time step of the parameterisations is increased to 10 min. The behaviour of the benchmark box model is shown in red which is still closely followed by the new parameterisation (blue stars). The parameterisation that uses saturation adjustment underestimates relative humidity during the cooling period and overestimates it during the warming period. Right panel: The corresponding mean in-cloud relative humidity for the stochastic model, the new parameterisation, and the parameterisation with saturation adjustment. The box model and the new parameterisation approach the equilibrium supersaturation (black dashed), while saturation adjustment assumes exactly saturation inside the cloud.

deviation in relative humidity is rather due to the rough approximation of the time dependence of the updraught resulting in a deviation in temperature. Finally, the time step was increased to 30 minutes and the deviation of the new parameterisation from the stochastic box model was still small (not shown).

Overall, the new parameterisation always stayed close to the stochastic box model in all performed tests and thus has proven to be numerically stable against variations of all considered parameters.

## 4 Discussion

It is important to see that saturation adjustment is a special case of the new parameterisation, namely one with a very large value of the deposition rate $\alpha$ (or equivalently a very short phase relaxation time). Our first sensitivity study (Figs. 8 and 9) has shown that both parameterisations give more similar results if the phase relaxation time is short. Eq. 11 shows that the equilibrium supersaturation approaches zero (that is, the relative humidity over ice approaches 100%) when $\alpha$ becomes very large. Thus, both causes for the underestimation of supersaturation vanish in this limit. However, this limit is not the typical case for cirrus clouds.

One can estimate a value for $\alpha$ as the reciprocal value of the phase relaxation time given by Khvorostyanov and Sassen (1998a):

$$\alpha = 4\pi\,D\,N\,\overline{r}, \tag{54}$$

where $D$ is the temperature and pressure dependent water vapour diffusion coefficient, $N$ is the number concentration of ice crystals and $\overline{r}$ is the radius that they would obtain if the complete excess vapour was transferred into $N$ spherical crystals. ~~is their mean radius.~~ In this equation, the diffusion coefficient has the smallest variability (Pruppacher and Klett, 1997), while the number density and crystal radius vary over several orders of magnitude (Dowling and Radke, 1990; Heymsfield et al., 2017; Krämer et al., 2009, 2016, 2020). The data in these papers indicate that short relaxation times (high $N$ and large crystals) are typical for liquid-origin cirrus, while in-situ formed cirrus is rather characterised by longer relaxation times and thus we can expect long lasting in-cloud supersaturation in this kind of cirrus. In contrast, liquid origin cirrus is characterised by its liquid origin, that is, it started from water saturation, which implies high ice supersaturation that relaxes to ice saturation quickly.

In the tests described above, we kept the value of $\alpha$ constant throughout each simulation. This is similar to the approach with saturation adjustment, where, although $\alpha$ is not specified, we can conceive it as a constant very large value. However, a constant $\alpha$ is not a condition for the new parameterisation and we plan to recalculate it every time step for every grid box based on the individual meteorological conditions influencing the quantities in eq. 54 when implementing the parameterisation into an actual 3D-NWP model. ~~We have tested this as well (results not shown). A value of $\alpha$ can in principle always be estimated in the course of a simulation.~~ First, $N$ can be estimated via the updraught speed, $w$, when the cloud forms, using the $N \propto w^{3/2}$ relation from Kärcher and Lohmann (2002). This relation only holds for clear air and a lower number density has to be assumed if pre-existing ice is present. Secondly, $\overline{r}$ can be estimated by assuming that the supersaturation is completely transferred to spherical ice crystals with radius $\overline{r}$. In a further test with the stochastic box model we let $\alpha$ vary with $w^{3/2}$ for every air parcel. For this test, we also introduced uniformly distributed, random sub-grid fluctuations in $w$ of $\pm 1 \, \mathrm{cm \, s^{-1}}$ to investigate how small scale updraught fluctuations affect the nucleation rate. Every second each air parcel therefore experienced a new random updraught between $1 \, \mathrm{cm \, s^{-1}}$ and $3 \, \mathrm{cm \, s^{-1}}$. Fig. 15 shows that the updraught fluctuations result in an overall higher mean value of $\alpha$ leading to a slightly lower relative humidity shortly after the onset of nucleation. However, the non-linear dependence of $S_{\mathrm{eq}}$ on $\alpha$ and $\frac{\mathrm{d \ln(q_s)}}{\mathrm{dt}}$ causes an increase of the equilibrium supersaturation, thereby leading to an increased relative humidity in the aged cloud. Overall, the two simulations do not differ very much which gives hope that the negligence of updraught fluctuations an the use of a single value for $\alpha$ do not introduce large errors into simulations of the new parameterisation. ~~If the same $\alpha[w(t)]$ applies to all air parcels in the statistical box model, the results of the new parameterisation and the box model are again nearly equal. Moderate differences appear, however, if we let every air parcel have its individual $\alpha$. This is obviously a quite complicated situation that can no longer be fully captured in a one-moment model.~~

Another process that has been simplified is the process of nucleation. On the one hand, recent studies have found that especially in slow updraughts the nucleation threshold from Kärcher and Lohmann (2002) is often not reached, as homogeneous nucleation already takes place at low rates when the supersaturation is still below the threshold (Baumgartner and Spichtinger, 2019; Spichtinger et al., 2023). If these low nucleation rates are active for longer times (as in a slow updraught), the amount of generated ice crystals becomes large enough to reduce the supersaturation without the actual nucleation threshold ever being reached. Thus, nucleation might actually occur earlier than in the presented simulations. However, we here stick with the original threshold from Kärcher and Lohmann (2002) as this is also the one used in the IFS. Switching to a more elaborated nucleation threshold involving not only supersaturation but also the vertical velocity should be fairly straight forward. On the

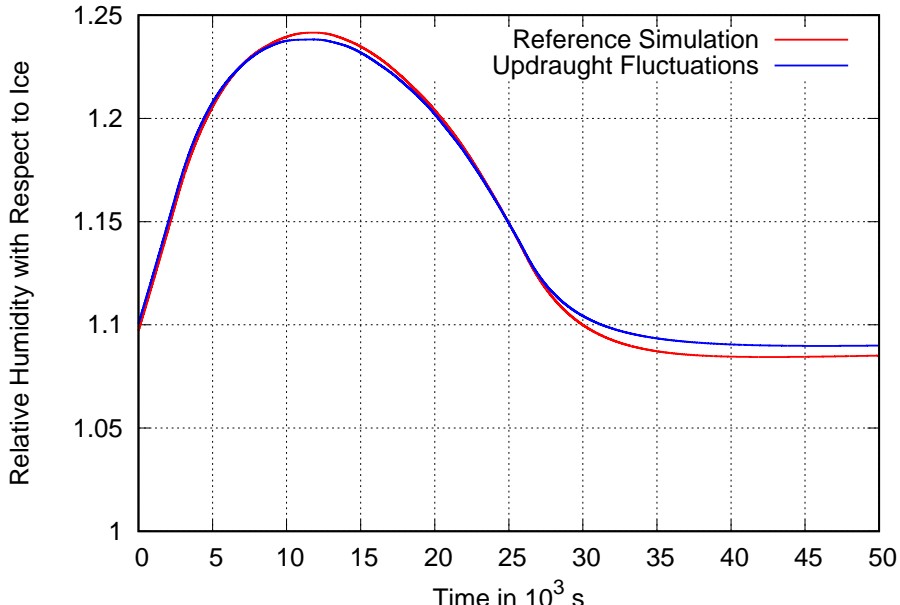

**Figure 15.** Evolution of the grid box mean relative humidity in the stochastic box model in a simulation, where every time step each air parcel experiences a random updraught between $1\,\mathrm{cm\,s^{-1}}$ and $3\,\mathrm{cm\,s^{-1}}$ (blue). $\alpha$ varies with the updraught velocity at the time of nucleation in the individual air parcel around $\alpha_0 = 3 \cdot 10^{-4}\,\mathrm{s^{-1}}$. The red plot shows the reference simulation of the stochastic box model from fig. 6 with uniform $w = 2\,\mathrm{cm\,s^{-1}}$ and uniform $\alpha = 3 \cdot 10^{-4}\,\mathrm{s^{-1}}$ for all air parcels. Even though the updraught fluctuations cause a slightly larger mean value of $\alpha$, they lead to an increased equilibrium humidity in the aged cloud due to the non-linear dependence of $S_\mathrm{eq}$ on $\frac{\mathrm{d\,ln(q_s)}}{\mathrm{dt}}$.

other hand, nucleation and initial crystal growth are not instantaneous in reality such that $\alpha$ would actually have to start from zero at the onset of nucleation and increase over time. Gierens (2003) provides a formulation of $\alpha$ that includes this effect in a simple exponential form which we implemented into our box model to evaluate this effect:

$$\alpha(t) = \alpha_0 \left( 1 - e^{-\alpha_0(t - t_\mathrm{nuc})} \right), \tag{55}$$

where $\alpha_0$ is the assumed final value of $\alpha$ after the initial crystal growth phase and $t_\mathrm{nuc}$ is the time at which nucleation started

in the respective air parcel. As can be seen in fig. 16 accounting for the duration of the nucleation event itself leads to even larger supersaturations and an even slower relaxation to the equilibrium supersaturation. However, we do not plan to adapt the new parameterisation to include this effect as it might become impossible to derive an analytical form of the in-cloud humidity distribution (eq. 17) which would increase the complexity and thus the run time of the parameterisation.

       Finally, we also neglected the process of heterogeneous nucleation so far. In order to investigate the influence of ice nucle-

ating particles (INP) on the evolution of relative humidity in our simulations, we consider the critical number density $N_\mathrm{cr}$ of INPs derived by Gierens (2003) (eq. 21 therein) which gives an approximate order of magnitude above which cloud formation is dominated by heterogeneous nucleation. Since Haag et al. (2003) suggested that even in the polluted regions of the northern hemisphere cirrus clouds do not form exclusively via heterogeneous nucleation, we take half of $N_\mathrm{cr}$ as our number density of

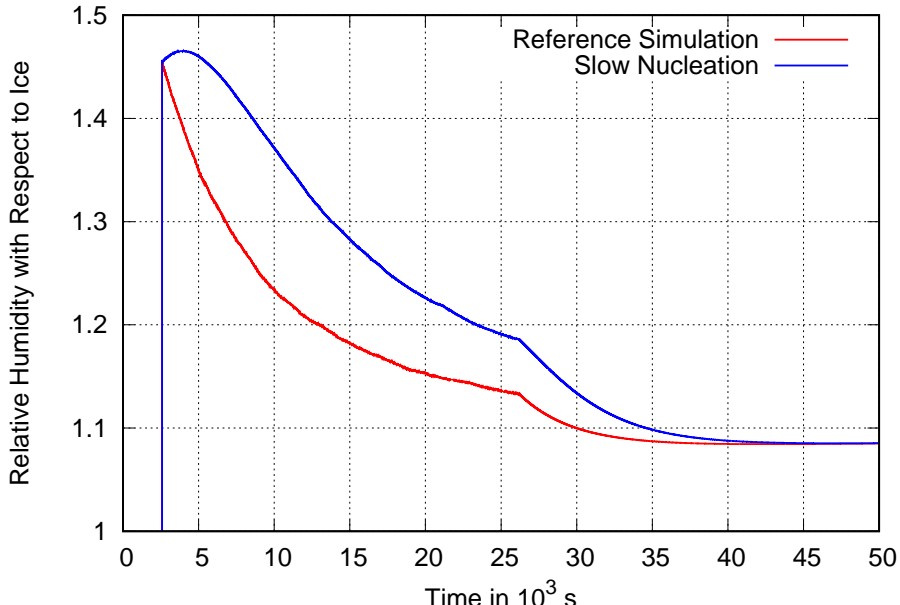

**Figure 16.** Evolution of the in-cloud mean relative humidity in the stochastic box model in a simulation, where $\alpha$ starts at zero at the onset of nucleation in every air parcel and increases over time to $\alpha = 3 \cdot 10^{-4}$ s$^{-1}$ (blue) compared to the reference simulation of the stochastic box model from fig. 6 with constant $\alpha = 3 \cdot 10^{-4}$ s$^{-1}$ (red). Accounting for the phase of initial crystal growth leads to a short further increase in supersaturation after the onset of nucleation before deposition becomes fast enough to deplete the supersaturation resulting in a considerable delay of the relaxation process.

INPs. We introduce it into our stochastic box model by setting a second nucleation threshold at $130\%$ relative humidity and
calculating a corresponding value of $\alpha$ via eq. 54 for deposition on the resulting heterogeneously nucleated ice particles. In particular, the number density of INPs is $128$ m$^{-3}$ and the corresponding deposition rate is $1.5 \cdot 10^{-5}$ s$^{-1}$. Consequently, if an air parcel's relative humidity exceeds $130\%$, deposition is assumed to occur at this low rate, while $\alpha$ returns to its usual value of $3 \cdot 10^{-4}$ s$^{-1}$ if the parcel's relative humidity exceeds the threshold for homogeneous nucleation. Fig. 17 shows the result of this simulation. Intuitively, one might think introducing another sink for water vapour should reduce the relative humidity.
However, apparently the opposite is the case. Heterogeneous nucleation removes moisture from the air parcels with the highest vapour content, while the increase in relative humidity in the dryer parcels remains unchanged. As a result, the threshold for homogeneous nucleation is reached later and the grid mean relative humidity increases longer leading to an overall increased grid mean relative humidity. Hence, the negligence of heterogeneous nucleation in the new parameterisation does not lead to an overestimation of relative humidity in situations as the one considered here. Obviously, the picture changes if the number den-
sity of INPs exceeds $N_{cr}$ by far such that the heterogeneously induced deposition becomes fast enough to prevent homogeneous nucleation from occurring. But according to Haag et al. (2003) such high number densities of INPs should be uncommon.

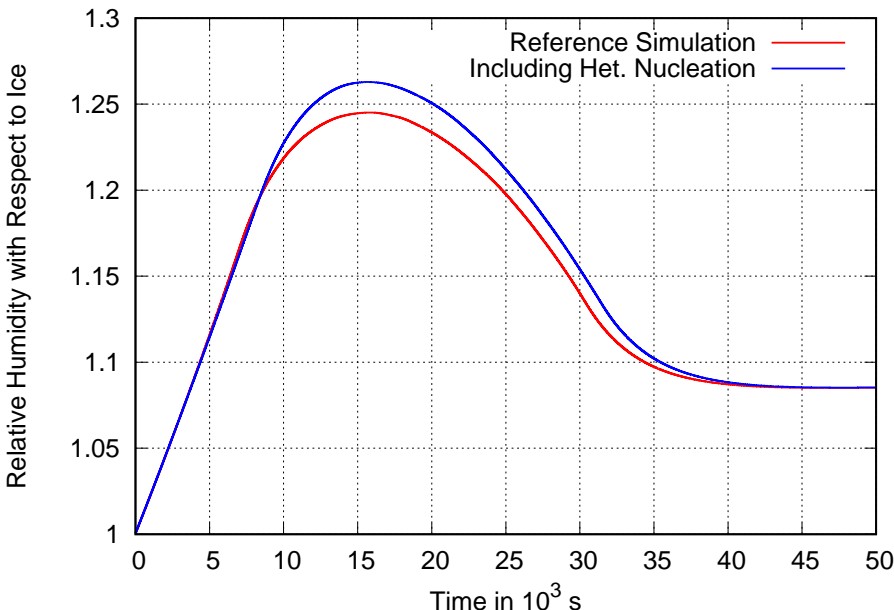

**Figure 17.** Evolution of the grid box mean relative humidity in the stochastic box model in a simulation with a second nucleation threshold at $130\%$ representing heterogeneous nucleation (blue). For each air parcel, $\alpha$ is set to $1.5 \cdot 10^{-5}$ s$^{-1}$ once its relative humidity exceeds $130\%$ and to $3 \cdot 10^{-4}$ s$^{-1}$ once it exceeds the threshold for homogeneous nucleation. The red plot shows the reference simulation of the stochastic box model from fig. 6 with constant $\alpha = 3 \cdot 10^{-4}$ s$^{-1}$ but starting from $100\%$ relative humidity. The heterogeneously induced deposition removes vapour from the moistest air parcels thereby delaying the onset of homogeneous nucleation. This leads to an overall increase in the grid box mean relative humidity.

Further complex influences on the value of $\alpha$ and thus on the distribution of supersaturation within cirrus clouds may arise due to advection and sedimentation processes which we have not considered so far in the development of the new concept. A model with saturation adjustment does not have such influences because its $\alpha$ is simply infinite and its distribution of in-cloud supersaturation a $\delta$-function without any sensitivity to whatever other microphysical processes. An investigation of the behaviour of the new parameterisation in a more complete framework that includes advection and sedimentation is out of scope for the present paper, but is of course the necessary next step in the implementation of the new concept into an actual NWP model.

The mentioned influences on $\alpha$ might be better captured in a two-moment scheme where information of ice crystal concentration is available. We expect that a combination of the new concept with explicit in-cloud supersaturation and a two-moment scheme will be fruitful.

In the end, even the best parameterisation will not lead to a temporally and spatially precise prediction of ice-supersaturation alone without the aid of relative humidity data for data assimilation, as argued in the introduction. The new concept can make the model humidity statistics better match the corresponding statistics from measurements taken over many years in the

MOZAIC (Measurement of ozone and water vapor by Airbus in-service aircraft, Marenco et al., 1998) and IAGOS projects (In-service Aircraft for a Global Observing System, Petzold et al., 2015); but this is only a necessary, not a sufficient condition. More aircraft equipped with hygrometers that work reliably in the upper troposphere are urgently needed.

Tompkins et al. (2007) noticed that many ice cloud parameterisations suffer from unphysical sub-grid humidity fluxes between the cloudy and the clear sky portion of a grid box. These fluxes can occur if the specific humidities inside the cloud and in the cloud environment are coupled to the grid mean humidity via a diagnostic assumption. Since our parameterisation uses the in-cloud humidity as an additional source of information, there is no need for a diagnostic assumption and unphysical sub-grid humidity fluxes cannot occur, as the in-cloud humidity is processed independently of the grid mean humidity.

Concerning the run time of the new parameterisation, 850 time steps took about 7 ms in the current version, while the parameterisation with saturation adjustment took about 4.5 ms for the same number of steps on the same computer. Furthermore, there will be an increase in run time of a numerical weather prediction model due to the additional prognostic variable that is needed in the new parameterisation. Hence, one has to expect additional computational costs when implementing the new parameterisation.

## 5 Summary and Conclusions

This paper has been written in response to a problem that is probably common to most NWP and climate models, namely the underestimation of both the frequency and degree of ice supersaturation in the upper troposphere. There are many ways that this problem can be tackled and mitigated, ranging from simple corrections to the output humidity field to elaborated cloud microphysics submodels that represent many processes, full size spectra of ice crystals and aerosols, and the complex dynamical background in which the clouds are embedded and in which the microphysics proceeds. Evidently, there is a trade-off between microphysics elaboration and computing effort, and our guideline here is to make better predictions of ice supersaturation in a model that is still cheap and thus fast enough that it can serve as a NWP model (which has to obey strict run-time constraints). Thus, we try to stay with a one-moment scheme, knowing that better but more expensive schemes exist, and we boil down the nucleation and crystal growth physics to a quite simple formulation which, however, is promising to provide a better (albeit not the best) representation of ice supersaturation in NWP models than traditional methods that use saturation adjustment. In the sense of "adequacy for purpose" (Gramelsberger et al., 2020; Parker, 2020) we think this is a good compromise.

In this study we have introduced the concept of a new parameterisation for the representation of pure ice clouds in numerical weather prediction models that overcomes the practice of saturation adjustment. This common practice that consists in assuming exactly saturated conditions inside a cloud has been compared to a stochastic box model in which the model grid box is divided into a large number of air parcels. Every air parcel can become cloudy individually if it meets the conditions for nucleation, in which case the humidity inside the parcel is assumed to exponentially decay towards saturation. Since this assumption can be considered as a reasonable approximation to reality, the mean humidity across all air parcels inside this box model serves as a benchmark to estimate the quality of the considered parameterisations.

In this comparison, the parameterisation using saturation adjustment has been shown to underestimate humidity in the presence of a cloud for two reasons:

- The large supersaturation at the onset of nucleation is converted immediately into ice; a process that can take up to hours in weak updraughts in reality.

- As long as cooling proceeds, it represents a continuous source of new supersaturation such that the cloud remains in a slightly supersaturated state that is not represented in saturation adjustment.

Furthermore, humidity can also be overestimated by saturation adjustment if the temperature inside the cloud is rising. In this case, the ice crystals need time to sublimate such that the air inside the cloud becomes slightly subsaturated, while saturation adjustment again assumes exactly saturated conditions.

This insufficient treatment of in-cloud humidity in current NWP models is one of the factors that hamper a reliable prediction of persistent contrails, which contribute significantly to the overall warming effect of aviation emissions. Our new parameterisation aims to better represent in-cloud humidity by introducing it as a new prognostic variable. We thus can explicitly model the decay of the initial, large supersaturation right from the onset of nucleation and also represent any super- or subsaturated state in the later life cycle of a cirrus cloud. For this purpose, we derived the humidity distribution in newly generated cloud parts from the stochastic box model.

The new parameterisation has been shown to closely follow the behaviour of the stochastic box model for different updraught scenarios, different rates of exponential humidity decay, different widths of the sub-grid humidity fluctuation distribution and different time step lengths. In particular, it came out to always be an improvement to saturation adjustment. This improvement is larger if cloud fraction increases fast, as this gives more weight to the initially highly supersaturated cloud in the grid box mean humidity, for example if the sub-grid humidity fluctuations in the clear sky are small. On the other hand, the improvement to saturation adjustment is small if a large number of ice crystals is generated upon nucleation inside the cloud, for example in a strong updraught. In this case, the in-cloud humidity relaxes towards saturation quickly and also the equilibrium supersaturation that it approaches is small, since the available surface for the deposition of water vapour is large. However, such conditions are not typical for synoptic scale cirrus and hence a significant improvement compared with a saturation adjustment scheme can be expected.

In reality, the rate of the exponential decay of humidity depends on the number density and the size of the ice crystals inside the cloud and thus varies in time. Although this rate has been held constant throughout each of the presented tests of the new parameterisation, first simulations with a variable decay rate have provided promising results. Of course, further tests, especially in a less artificial environment, are needed but we are confident that, despite the additional computational costs, our parameterisation is capable of contributing to a significant improvement of the humidity forecast in the upper troposphere.

## Appendix A: Derivation of the Equilibrium Supersaturation

Here we provide details on the derivation of $S_{\mathrm{eq}}$. The defining condition for the equilibrium supersaturation is that a parcel's supersaturation $S_p$ remains unchanged if it equals $S_{\mathrm{eq}}$:

$$\left(\frac{\mathrm{dS_p}}{\mathrm{dt}}\right)_{\mathrm{eq}} = \frac{\mathrm{d}}{\mathrm{dt}}\left(\frac{q_p}{q_s} - 1\right)_{\mathrm{eq}} = 0 \tag{A1}$$

Using the product rule leads to

$$\frac{1}{q_s}\frac{\mathrm{dq_p}}{\mathrm{dt}} + q_p\frac{\mathrm{d}}{\mathrm{dt}}\left(\frac{1}{q_s}\right) = \frac{1}{q_s}\frac{\mathrm{dq_p}}{\mathrm{dt}} - \frac{q_p}{q_s^2}\frac{\mathrm{dq_s}}{\mathrm{dt}} = 0. \tag{A2}$$

By moving the second term to the right side of the equation and multiplying it with $q_s$ and dividing by $q_p$, we get

$$\frac{1}{q_p}\frac{\mathrm{dq_p}}{\mathrm{dt}} = \frac{1}{q_s}\frac{\mathrm{dq_s}}{\mathrm{dt}} \tag{A3}$$

$$\frac{\mathrm{d\ln q_p}}{\mathrm{dt}} = \frac{\mathrm{d\ln q_s}}{\mathrm{dt}}, \tag{A4}$$

which means that in order for $S_p$ to remain constant, the relative decrease rates of the parcel's specific humidity due to deposition and the saturation specific humidity due to cooling have to equal each other. We now use eq. 3 to further evolve the expression:

$$-\frac{1}{q_p}\alpha(q_p - q_s) = \frac{\mathrm{d\ln q_s}}{\mathrm{dt}} \tag{A5}$$

After multiplying this equation with $q_p$ and dividing by $\alpha$ and $q_s$, we can identify the equilibrium supersaturation on the left side of the equation and the equilibrium relative humidity which can be written as $S_{\mathrm{eq}} + 1$ on the right side:

$$\frac{q_p - q_s}{q_s} = -\frac{1}{\alpha}\frac{q_p}{q_s}\frac{\mathrm{d\ln q_s}}{\mathrm{dt}} \tag{A6}$$

$$S_{\mathrm{eq}} = -\frac{1}{\alpha}(S_{\mathrm{eq}} + 1)\frac{\mathrm{d\ln q_s}}{\mathrm{dt}} \tag{A7}$$

Dividing by $S_{\mathrm{eq}} + 1$ and taking the reciprocal gives

$$\frac{S_{\mathrm{eq}}}{S_{\mathrm{eq}} + 1} = -\frac{1}{\alpha}\frac{\mathrm{d\ln q_s}}{\mathrm{dt}} \tag{A8}$$

$$1 + \frac{1}{S_{\mathrm{eq}}} = -\frac{\alpha}{\mathrm{d\ln q_s}/\mathrm{dt}}, \tag{A9}$$

where subtracting one and again taking the reciprocal leads to the desired result

$$S_{\mathrm{eq}} = \left(\frac{\alpha}{-\mathrm{d\ln q_s}/\mathrm{dt}} - 1\right)^{-1}. \tag{A10}$$

*Data availability.* Simulation data are available at doi: 10.5281/zenodo.8413820

*Author contributions.* This paper is part of Dario Sperber's master thesis. KG made the concept of the research, DS wrote the codes, ran the simulations and produced the figures. Both authors discussed the results and wrote the paper.

*Competing interests.* The authors declare no competing interests.

*Acknowledgements.* The research described in this paper contributes to the Horizon 2020 project BeCoM (Better Contrail Mitigation, Grant No. 101056885). The authors would like to thank Sabine Brinkop for her helpful comments and grant her a jar of tea in deep gratitude. Two
reviewers helped a lot to raise the quality of the paper with their insightful comments, for which we are grateful.

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
