# Peer review of "Towards a more reliable forecast of ice supersaturation: Concept of a one-moment ice cloud scheme that avoids saturation adjustment"

_EGUsphere, 2023_

## Referee Comment (RC2)

Review of

**Towards a more reliable forecast of ice supersaturation: Concept of a one-moment ice cloud scheme that avoids saturation adjustment**

by Sperber & Gierens

**Summary and general comment:**
In this study a new concept of handling ice supersaturation in a one-moment scheme is proposed. The model is based on a stochastic approach, starting with a stochastic box model which is then transferred to a grid box of a hypothetical coarse resolution model. The authors explain the theoretic concept, derive the relevant equations for the representation of ice supersaturation in a grid box and run some simulations for checking the agreement between the different models. Overall, there is a potential for representing ice supersaturation in a better way as compared to the instantaneous relaxation of in cloud water vapor to saturation.

The model is based on a very innovative and interesting idea, which leads to a really non-standard approach for representing ice supersaturation in simple one moment schemes. The study is conceptually new and thus is very well suited for ACP.

However, I had a hard time to understand the derivation of the equations and the details of the model, since they are often non-standard and are quite compactly represented. Thus, this very interesting paper could be improved in terms of the representation. In addition, for some scenarios the model has some issues to represent the physical behavior, which could be clarified. Therefore, I would recommend (major) revisions of the manuscript, before the study can be accepted for publication. In the following I will explain my concerns in details.

**Major issues**

1. Representation of the nucleation event:
   In the stochastic model, nucleation takes place instantaneously if the nucleation threshold is reached. There are some issues with this approach. First, the nucleation threshold as derived by Kärcher & Lohmann (2002) is probably not the right quantity for initiating the nucleation, since it will often not be reached (see Spichtinger et al., 2023), especially not during very low vertical velocities/cooling rates, as used for the simulations. Generally, ice nucleation is already triggered if the saturation ratio is close enough to the threshold, see discussion in Baumgartner & Spichtinger (2019). Using the high threshold by Kärcher & Lohmann (2002) might introduce an (incorrect) time shift. Second, the nucleation event itself has a duration, i.e. one has to wait until the nucleation event is completed and the full number of ice crystals (as e.g. diagnosed by the Kärcher & Lohmann scheme) is produced. This time is not included in the model, and might also lead to a (correct) time shift in the cloud evolution; to be precise, the small cap (or parabola) around the maximum in $s$ is missing (see, e.g., figure 1 in Spichtinger & Krämer, 2013). Although it is probably not possible to include the duration of the nucleation event into the model, this should be commented in the text.

2. Notation and derivation of some key equations:
   The key quantities as supersaturation $s$ and others should be introduced more carefully, since different communities use rather saturation ratio than supersaturation. Actually, there is no definition of $s$ in the manuscript.

   I failed to reproduce equation (9) from equations (3) and (8). It would be good if the derivation would be a bit more elaborated; maybe some details can be provided in an appendix. In this context, it would be good to have a formula for the key value $\alpha$ instead of referencing the quite confusing papers by Khvorostyanov & Sassen (1998a,b). In the end of the text there is a kind of explanation, but it would be good to have such a formula at the very beginning, when the key equation (3) is stated. Finally, it would be good to explain carefully, that the equilibrium supersaturation is depending on the cooling rate and can be positive and negative (saturation ratio above and below 1, see figures 3 and following). In the plots, it is also a bit confusing that RHi is used (but as saturation ratio), although in the text and the derivation the supersaturation is used.

Finally, it would enhance the readability if the notation $\exp(x)$ would be used instead of $e^x$. Especially, for equations (23), (35) etc. this would help a lot.

3. Use of a constant $\alpha$:
For slow updraft/cooling regimes the model seems to work quite well, but the authors report that there are issues for higher cooling rates. Especially, the equilibrium supersaturation becomes very high and a relaxation is difficult. To my opinion this stems (at least partially) from the fact that the value $\alpha$ is held constant, although it is composed by number concentration $n$ (which is constant after nucleation) and mean radius $\bar{r}$, which is NOT constant. Thus, the relaxation should be different from exponential (faster or slower) due to the additional change in $\alpha$. Maybe this change in $\alpha$ can be introduced into the model by an additional integration of the mass growth equation for a single crystal, because a monodisperse distribution is used anyway. However, the authors should check, if a change in the mean radius might improve their model results. This should be included into the discussion.

**Minor issues:**

1. Thermodynamic states:
In the abstract, ice clouds in supersaturated air are mentioned, but they can also survive in (moderately) subsaturated air. I would rather use "to be out of equilibrium" instead of "in an ice supersaturated state". Later in the text, ice supersaturation is termed to be an extreme case; maybe again the notation of "far away from equilibrium" might be more appropriate.

2. Contrail radiative effects:
I missed the reference Stuber et al. (2006), which clearly indicated the net warming of contrails due to the infrared component of the radiation.

3. Measurements:
In the introduction, the lack of measurements is strongly emphasized; however, in the discussion, the long term programs MOZAIC and IAGOS are mentioned. Actually, I missed these references in the introduction and would recommend to indicate that there are some (only sparse) measurements, which are indeed helpful.

4. Wording:
To my opinion, the term "saturation adjustment" is already taken; it is dedicated to a certain technique for modeling liquid clouds, although this procedure is closely related to the approach of fast relaxation of supersaturation to equilibrium. For warm clouds, at values of the saturation ratio above 1 (i.e. at very small supersaturation of order of few percents or even less), cloud droplets are formed instantaneously, the excess water vapor is transferred to cloud water, and the partial water vapor is set to saturation values. The principle for ice clouds allowing high supersaturations (of order of few 10 percent) before nucleation and then using a fast relaxation to equilibrium is similar, however a bit different to the original term (also without the numerical issues of latent heat release). Generally, I would like to avoid the term "saturation adjustment" in this context and rather use something like "fast relaxation", but this is a subjective viewpoint. However, I would ask the authors to clarify the term in context of the different thermodynamic phases (liquid clouds and ice clouds), since readers with a different background might be confused. Actually, there are also attempts to avoid saturation adjustment in liquid clouds allowing (moderate) supersaturations, see, e.g., Porz et al. (2018).

5. Figures and figure captions:
For figures 3 and 5 (and others of the constant updraft scenario) I would like to see the evolution of the cloud fraction; this would help to interpret the change in RHi as indicated in the caption. For figures 4 and 6 (and others of a variable updraft scenario) an additional representation of the cooling rate (or updraft) would be helpful. More complete figure captions (for all figures) would also be helpful, e.g. just a remark "scanerio 2 but with a different whatever" does not really help to understand the figure. It would be good if the figures together with the caption could be understood without reading the text carefully.

6. Diagnostic relation for ice crystal number concentrations:
It is claimed that the relation $n \sim w^{\frac{3}{2}}$ by Kärcher & Lohmann (2002) should be used for deriving the ice crystal number concentration. However, one should mention here that the relation only works for clear air; for pre-existing ice, a reduction of the number concentration must be taken into account. This should be mentioned in the text, maybe also in the context of competing nucleation pathways, e.g. homogeneous and heterogeneous nucleation.

**References**

Porz, N., M. Hanke, M. Baumgartner, and P. Spichtinger, 2018: A model for warm clouds with implicit droplet activation, avoiding saturation adjustment, Math. Clim. Weather Forecast., 4, 50-78, doi: 10.1515/mcwf-2018-0003

Spichtinger, P. and M. Krämer, 2013: Tropical tropopause ice clouds: a dynamical approach to the mystery of low crystal numbers. Atmos. Chem. Phys., 13, 9801-9818, doi:10.5194/acp-13-9801-2013

Spichtinger, P., P. Marschalik, M. Baumgartner, 2023: Impact of formulations of the homogeneous nucleation rate on ice nucleation events in cirrus. Atmos. Chem. Phys., 23, 2035-2060, doi: 10.5194/acp-23-2035-2023

Stuber, N., Forster, P., Rädel, G., Shine, K., 2006: The importance of the diurnal and annual cycle of air traffic for contrail radiative forcing. Nature, 441, 864-867, doi:10.1038/nature04877

---

## Author Comment (AC1)

*The authors would like to thank the reviewers for their critical and helpful comments.*

*A few general statements (This paragraph is now also included in the paper at the beginning of the last section.)*

*This paper has been written in response to a problem that is probably common to most NWP and climate models, namely the underestimation of both the frequency and degree of ice supersaturation in the upper troposphere. There are many ways that this problem can be tackled and mitigated, ranging from simple corrections to the output humidity field to elaborated cloud microphysics submodels that represent many processes, full size spectra of ice crystals and aerosols, and the complex dynamical background in which the clouds are embedded and in which the microphysics proceeds. Evidently, there is a trade-off between microphysics elaboration and computing effort, and our guideline here is to make better predictions of ice supersaturation in a model that is still cheap and thus fast enough that it can serve as an NWP model (which has to obey strict run-time constraints). Thus, we try to stay with a one-moment scheme, knowing that better but more expensive schemes exist, and we boil down the nucleation and crystal growth physics to a quite simple formulation which, however, is promising to provide a better (albeit not the best) representation of ice supersaturation in NWP models than traditional methods that use saturation adjustment. In the sense of "adequacy for purpose" (Parker, 2020; Gramelsberger et al., 2020) we think this is a good compromise.*

*In the following our answers to the individual comments are written in italic letters.*

**Comments of Blaž Gasparini:**

The manuscript by Sperber and Gierens, 2023 introduces a new cirrus parameterization that avoids saturation adjustment and can be implemented in models using one-moment microphysical schemes. The scheme adds an additional prognostic variable describing the in-cloud humidity to the reference model using the saturation adjustment assumption. The authors describe and compare the results of their scheme with the benchmark stochastic model and the reference saturation-adjusted cloud model, demonstrating the advantage of their parameterization.

This is a valuable study that could potentially in the long-term lead to improvements in coarse-resolution models using simple, 1-moment microphysical schemes and the saturation adjustment assumption. Such simplified representations of cirrus are common in current weather prediction models. Overall, it is a very nice parameterization development study that has some potential for improving ice supersaturation forecasts. However, I have a number of questions that should be addressed before the study can be published in its final form.

General comments:

The conditions where the new parameterization improves the modeled moisture fields the most (slow, constant updraft) are most likely to be associated with heterogeneous freezing at lower supersaturations. If this is the case, and nucleation occurs at lower supersaturations, the advantage of your scheme would probably be smaller compared to the cases presented. Also, even if your scheme removes one of the biases, bias due to the lack of heterogeneous ice nucleation may persist.

I am wondering which of the two biases could have larger implications on the simulated supersaturation fields, particularly in the context of a weather model. Please, discuss!

*REPLY: Good point! We have now made some tests with the stochastic model to see what happens. The results of these tests are included and described in the discussion section of the revised manuscript.*

Also, I think it might be worthwhile to add an additional sensitivity test assuming an idealized type of heterogeneous ice nucleation. This test might give some more clues about the importance of the new parameterization.

*REPLY: Yes, this has now been done using the stochastic model. We simply introduce a second nucleation threshold and let $\alpha$ have two values, a smaller one for the het. nucleation (small number of ice crystals) and a larger one for homogeneous nucleation (many ice crystals). The value of $\alpha$ switches as soon as an air parcel reaches the threshold for homogeneous nucleation.*

Indeed, I know that weather models typically don't have either interactive or prescribed aerosol fields that could modify cirrus formation. However, prescribing an aerosol or ice nucleating particle climatology would be a fairly straightforward task.

*REPLY: The value of $\alpha$ depends on the number of nucleated ice crystals, thus indirectly on the number of nucleating particles for both heterogeneous and homogeneous nucleation. Thus, a climatology or else of nucleating particles is not necessary in our formulation. In an actual application such a climatology would of course help to specify the value of $\alpha$, which here is simply a free parameter.*

How realistic is the assumption of a constant updraft for periods longer than 12 hours? Does this really occur in the real atmosphere? How well are such processes simulated by weather models?

A number of studies point to the importance of rapid temperature/updraft variations in the formation of cirrus, particularly in the context of homogeneous freezing. The manuscript mentions the inclusion of such variability in the scheme. However, I didn't understand how such fluctuations are included in the new scheme. Could you explain this better? What if the variability were larger/smaller?

*REPLY: Of course, a constant (in the mathematical sense) updraft is an idealisation. A 12 hours "constant" updraft might occur related to synoptic systems (with time scales of a couple of days), e.g. when the warm air is flowing above a warm front. But of course, there are always fluctuations of vertical windspeed, which we haven't considered so far, since our main goal was not to represent reality as closely as possible, but to set up a concept for a cirrus parameterisation that better represents ice supersaturation. Nevertheless, the extended discussion section now includes a*

*simulation of the stochastic model with rapid fluctuations in w. However, this is so far only for discussion. We do not plan to make our concept and the parametrisation more complex to also represent such, perhaps minor, effects. This is left for future model developments, if somebody deems it is necessary for certain purposes. Our main priority for the further development is the implementation of the concept into a real NWP-model, where the concept has to proof its usefulness in interaction with all other processes.*

I believe the authors hate to provide a data availability statement, as per https://www.atmospheric-chemistry-and-physics.net/policies/data_policy.html

*REPLY: Thanks, a data availability statement is now included.*

Specific comments:

Page 1, abstract: I'm missing a sentence on the conditions in which the new scheme is improving humidity most compared to saturation adjustment

*REPLY: Such a sentence is included now in the revised version.*

Page 2, first paragraph: I like the sharp focus on contrails. However, from the climatic point of view, the scheme would have likely a much larger impact on natural cirrus coverage and their radiative effects.

*REPLY: This is certainly true. If this were a concept for a climate model, it would require new tuning. For an NWP model this is probably not necessary because of the data assimilation which keeps the forecast close to reality. We think also (or hope) that a change in cirrus coverage and radiative effects will not have too large an effect on predicted surface weather, since up to now a realistic treatment of cirrus turned out not to be critical for surface weather forecast.*

Page 5, equation 10: Why is the assumption of a hyperbolic supersaturation? Where does this come from?

*REPLY: This is the result of the transformation of a uniform pdf of nucleation times into the supersaturation pdf for cloudy air parcels (transformation of probability densities). The revised version will give more details as also the 2$^{nd}$ reviewer asked for them.*

Page 6: Why can't growing ice crystals decrease the relative humidity to 100%?

*REPLY: As long as there is uplift, the supersaturation is steadily renewed (equilibrium between sink and source). In order to reach exactly 100%, the uplift (and thus the cooling) must stop, that is, the temperature must no longer change.*

Section 2.2: please note that I did not go through the equations, but I just tried to qualitatively understand the concept of the new scheme. An illustration/schematic could help to effectively convey the key concept of the new scheme.

*REPLY: Yes, we understand this. A sketch will be included.*

Section 3: Visualizing the updrafts considered in the considered cases would improve the clarity of the results, particularly for the cosine-like cases.

*REPLY: Agreed.*

Figure 4 and 8: based on the caption I expect an increase in updraft from 2 to 5 cm/s. Please, just plot the updraft.

*REPLY: Agreed.*

Page 18, lines 364-370: Would be great to see that in a separate sensitivity test and a new figure.

*REPLY: Agreed, see additional experiment with fluctuations in w.*

Page 19, lines 404-405: language, please rewrite the sentence. Saturation adjustment => parameterization using saturation adjustment (or similar)

*REPLY: Agreed.*

Page 19, line 408: "keeps getting restored" => is that true? I guess it's simply continuous?

*REPLY: Unclear, what you mean. Is restoration necessarily unsteady? Can't it be continuous? Perhaps we should write "keeps getting restored continuously".*

Page 19, line 426-427: suggestion "…a significant improvement compared with a saturation adjustment scheme…"

*REPLY: Thanks for the suggestion which we follow.*

Citation: https://doi.org/10.5194/egusphere-2023-914-RC1

**Comments of Referee #2:**

Review of

**Towards a more reliable forecast of ice supersaturation: Concept of a one-moment ice cloud scheme that avoids saturation adjustment**

by Sperber & Gierens

**Summary and general comment:**

In this study a new concept of handling ice supersaturation in a one-moment scheme is proposed. The model is based on a stochastic approach, starting with a stochastic box model which is then transferred to a grid box of a hypothetical coarse resolution model. The authors explain the theoretic concept, derive the relevant equations for the representation of ice supersaturation in a grid box and run some simulations for checking the agreement between the different models. Overall, there is a potential for representing ice supersaturation in a better way as compared to the instantaneous relaxation of in cloud water vapor to saturation.

The model is based on a very innovative and interesting idea, which leads to a really non-standard approach for representing ice supersaturation in simple one moment schemes. The study is conceptually new and thus is very well suited for ACP.

However, I had a hard time to understand the derivation of the equations and the details of the model, since they are often non-standard and are quite compactly represented. Thus, this very interesting paper could be improved in terms of the representation. In addition, for some scenarios the model has some issues to represent the physical behavior, which could be clarified. Therefore, I would recommend (major) revisions of the manuscript, before the study can be accepted for publication. In the following I will explain my concerns in details.

**Major issues**

1. Representation of the nucleation event:
   In the stochastic model, nucleation takes place instantaneously if the nucleation threshold is reached. There are some issues with this approach. First, the nucleation threshold as derived by Kärcher & Lohmann (2002) is probably not the right quantity for initiating the nucleation, since it will often not be reached (see Spichtinger et al., 2023), especially not during very low vertical velocities/cooling rates, as used for the simulations. Generally, ice nucleation is already triggered if the saturation ratio is close enough to the threshold, see discussion in Baumgartner & Spichtinger (2019). Using the high threshold by Kärcher & Lohmann (2002) might introduce an (incorrect) time shift. Second, the nucleation event itself has a duration, i.e. one has to wait until the nucleation event is completed and the full number of ice crystals (as e.g. diagnosed by the Kärcher & Lohmann scheme) is produced. This time is not included

in the model, and might also lead to a (correct) time shift in the cloud evolution; to be precise, the small cap (or parabola) around the maximum in s is missing (see, e.g., figure 1 in Spichtinger & Krämer, 2013). Although it is probably not possible to include the duration of the nucleation event into the model, this should be commented in the text.

*REPLY: Let us divide the reply into two points according to the list of issues in the comment. 1) We use the threshold formulation of Kärcher and Lohmann (2002) because it is implemented in the Tompkins et al. IFS-scheme, which is our point of reference for the concept development. We know that other formulations of the threshold are indeed possible, and for instance Gierens (2003) derived a slightly different one (Eq. 15) from his box model simulations, which has lower threshold values at T>230 K and higher threshold values at T<230 K. For the concept it does not play a role which threshold is used, and it would be a simple switch to another formulation in order to consider, for instance, the mentioned results by Baumgartner and Spichtinger (2019) and Spichtinger et al. (2023).*
*We mention this issue now in the enhanced discussion section.*
*2) The fact that nucleation starts slowly and not instantaneously is ignored in our concept for the sake of simplicity. Gierens (2003) uses a formulation that takes this effect into account and we have tried this (with a corresponding modification in the formulation of $\alpha$) in our stochastic model. Indeed, we get even higher in-cloud and grid-average supersaturations initially. This is now mentioned as well in the discussion section. However, for the formulation of the concept we don't plan to include this effect in order to keep it as simple as possible. The reason for this is that we would probably get a different pdf of the in-cloud supersaturation. It is not clear whether a simple analytical form would still be possible, and if this were not the case, the scheme would require a numeric integration in every time step to calculate the mean in-cloud supersaturation. This would need more run-time for every time step, hence we are reluctant with making the scheme more complex than absolutely necessary.*

2. Notation and derivation of some key equations:
   The key quantities as supersaturation s and others should be introduced more carefully, since different communities use rather saturation ratio than supersaturation. Actually, there is no definition of s in the manuscript.
   I failed to reproduce equation (9) from equations (3) and (8). It would be good if the derivation would be a bit more elaborated; maybe some details can be provided in an appendix. In this context, it would be good to have a formula for the key value α instead of referencing the quite confusing papers by Khvorostyanov & Sassen (1998a,b). In the end of the text there is a kind of explanation, but it would be good to have such a formula at the very beginning, when the key equation (3) is stated. Finally, it would be good to explain carefully, that the equilibrium supersaturation is depending on the cooling rate and can be positive and negative (saturation ratio above and below 1, see figures 3 and following). In the plots, it is also a bit confusing that RHi is used (but as saturation ratio), although in the text and the derivation the supersaturation is used. Finally, it would enhance the readability if the notation exp(x) would be used instead of $e^x$. Especially, for equations (23), (35) etc. this would help a lot.

*REPLY: Thanks for the comment. As the other reviewer likes to have more detail as well, we add an appendix to the revised version with the desired derivation. Otherwise, we try to clarify our formulation. See the revised manuscript.*

3. Use of a constant α:
   For slow updraft/cooling regimes the model seems to work quite well, but the authors report that there are issues for higher cooling rates. Especially, the equilibrium supersaturation becomes very high and a relaxation is difficult. To my opinion this stems (at least partially) from the fact that the value α is held constant, although it is composed by number concentration n (which is constant after nucleation) and mean radius $\bar{r}$, which is NOT constant. Thus, the relaxation should be different from exponential (faster or slower) due to the additional change in α. Maybe this change in α can be introduced into the model by an additional integration of the mass growth equation for a single crystal, because a monodisperse distribution is used anyway. However, the authors should check, if a change in the mean radius might improve their model results. This should be included into the discussion.

   *REPLY: Let us first clarify that $\alpha$, as introduced in our paper, is indeed constant and there is no inconsistency with the growing of the ice crystals. $\alpha$ depends on crystal number density N and on the radius the crystals would attain if the water vapour in excess of saturation was completely transformed into N spherical ice crystals (see Gierens, 2003, after eq. 7 and eq. 17). This may sound a bit strange, but it can be understood from the fact, that $\alpha$ is just the inverse of the time scale for relaxation of supersaturation. The definition of time scales has a certain degree of freedom; it just describes a "typical" time, not an exact duration. Consider the following example: The growth of ice crystals can be taken as growth of mass (volume), surface area, or radius. The physical process is the same but the time scales (say e-folding time) are not. These time scales differ by small factors, but for the statement of a "typical" duration this is good enough.*
   *It is certainly correct that the rate of deposition changes with the size of the ice crystals. We plan to recalculate α every time step based on the individual meteorological conditions in a future application of the parameterisation when it will be implemented in a 3D weather model and this will be noted in the discussion section. A changing α will not affect the decay within one time step but will of course change the overall temporal behaviour of in-cloud supersaturation across many time steps.*

**Minor issues:**

1. Thermodynamic states:
   In the abstract, ice clouds in supersaturated air are mentioned, but they can also survive in (moderately) subsaturated air. I would rather use "to be out of equilibrium" instead of "in an ice supersaturated state". Later in the text, ice supersaturation is termed to be an extreme case; maybe again the notation of "far away from equilibrium" might be more appropriate.

   *REPLY: Agreed.*

2. Contrail radiative effects:
   I missed the reference Stuber et al. (2006), which clearly indicated the net warming of contrails due to the infrared component of the radiation.

   *REPLY: Thanks, the reference will be included.*

3. Measurements:

In the introduction, the lack of measurements is strongly emphasized; however, in the discussion, the long term programs MOZAIC and IAGOS are mentioned. Actually, I missed these references in the introduction and would recommend to indicate that there are some (only sparse) measurements, which are indeed helpful.

*REPLY: Agreed.*

4. Wording:
To my opinion, the term "saturation adjustment" is already taken; it is dedicated to a certain technique for modeling liquid clouds, although this procedure is closely related to the approach of fast relaxation of supersaturation to equilibrium. For warm clouds, at values of the saturation ratio above 1 (i.e. at very small supersaturation of order of few percents or even less), cloud droplets are formed instantaneously, the excess water vapor is transferred to cloud water, and the partial water vapor is set to saturation values. The principle for ice clouds allowing high supersaturations (of order of few 10 percent) before nucleation and then using a fast relaxation to equilibrium is similar, however a bit different to the original term (also without the numerical issues of latent heat release). Generally, I would like to avoid the term "saturation adjustment" in this context and rather use something like "fast relaxation", but this is a subjective viewpoint. However, I would ask the authors to clarify the term in context of the different thermodynamic phases (liquid clouds and ice clouds), since readers with a different background might be confused. Actually, there are also attempts to avoid saturation adjustment in liquid clouds allowing (moderate) supersaturations, see, e.g., Porz et al. (2018).

*REPLY: Tompkins et al. (2007) and also the IFS-documentation which are two of our references write about an adjustment to saturated conditions which is why we think that the term "saturation adjustment" describes this practice in a very good way. We also think that "adjustment" describes what is done more honestly than for instance "relaxation". Adjustment clearly signals a manipulation or a trick in the model, while relaxation is physics and using this word would rather imply that what was done so far was indeed representing physics. However, we now introduce the term more carefully.*

5. Figures and figure captions:
For figures 3 and 5 (and others of the constant updraft scenario) I would like to see the evolution of the cloud fraction; this would help to interpret the change in RHi as indicated in the caption. For figures 4 and 6 (and others of a variable updraft scenario) an additional representation of the cooling rate (or updraft) would be helpful. More complete figure captions (for all figures) would also be helpful, e.g. just a remark "scanerio 2 but with a different whatever" does not really help to understand the figure. It would be good if the figures together with the caption could be understood without reading the text carefully.

*REPLY: Additional figures and extended captions will be included.*

6. Diagnostic relation for ice crystal number concentrations:
It is claimed that the relation $n \sim w^{3/2}$ by Kärcher & Lohmann (2002) should be used for deriving the ice crystal number concentration. However, one should mention here that the relation only works for clear air; for pre-existing ice, a reduction of the number concentration must be taken into account. This should be mentioned in the text, maybe also in the context of competing nucleation pathways, e.g. homogeneous and heterogeneous nucleation.

*REPLY: A comment on this will be added to the discussion.*

**References**

Porz, N., M. Hanke, M. Baumgartner, and P. Spichtinger, 2018: A model for warm clouds with implicit droplet activation, avoiding saturation adjustment, Math. Clim. Weather Forecast., 4, 50-78, doi: 10.1515/mcwf-2018-0003

Spichtinger, P. and M. Krämer, 2013: Tropical tropopause ice clouds: a dynamical approach to the mystery of low crystal numbers. Atmos. Chem. Phys., 13, 9801-9818, doi:10.5194/acp-13-9801-2013

Spichtinger, P., P. Marschalik, M. Baumgartner, 2023: Impact of formulations of the homogeneous nucleation rate on ice nucleation events in cirrus. Atmos. Chem. Phys., 23, 2035-2060, doi: 10.5194/acp23-2035-2023

Stuber, N., Forster, P., Rädel, G., Shine, K., 2006: The importance of the diurnal and annual cycle of air traffic for contrail radiative forcing. Nature, 441, 864-867, doi:10.1038/nature04877 3